# Two-dimensional amorphous NiO as a plasmonic photocatalyst for solar H$_2$ evolution

Zhaoyong Lin [1], Chun Du[1], Bo Yan[1], Chengxin Wang[1] & Guowei Yang[1]

Amorphous materials are usually evaluated as photocatalytically inactive due to the amorphous nature-induced self-trapping of tail states, in spite of their achievements in electrochemistry. NiO crystals fail to act as an individual reactor for photocatalytic H$_2$ evolution because of the intrinsic hole doping, regardless of their impressive cocatalytic ability for proton/electron transfer. Here we demonstrate that two-dimensional amorphous NiO nanostructure can act as an efficient and robust photocatalyst for solar H$_2$ evolution without any cocatalysts. Further, the antenna effect of surface plasmon resonance can be introduced to construct an incorporate antenna-reactor structure by increasing the electron doping. The solar H$_2$ evolution rate is improved by a factor of 19.4 through the surface plasmon resonance-mediated charge releasing. These findings thus open a door to applications of two-dimensional amorphous NiO as an advanced photocatalyst.

[1] State Key Laboratory of Optoelectronic Materials and Technologies, Nanotechnology Research Center, School of Materials Science & Engineering, Sun Yat-sen University, Guangzhou 510275 Guangdong, China. Correspondence and requests for materials should be addressed to G.Y. (email: stsygw@mail.sysu.edu.cn)

Photocatalysis is a promising technology to turn low-density solar energy into high-density chemical energy for fuel production and environment purification[1]. NiO, a low-cost and earth-abundant semiconductor, has been widely used as a cocatalyst in photocatalytic $H_2$ evolution due to the strong bonds to atomic hydrogen[2,3]. Nevertheless, it is intrinsically hole-doped with a very low ratio of free carriers to trapped holes, leading to the inappropriateness as a photocatalyst to supply electrons for the proton reduction[4].

Photocatalysts have so far focused on crystals since Fujishima and Honda realized water splitting over a $TiO_2$ single crystal in 1972, whereas amorphous materials are usually evaluated to be photocatalytically inactive[5]. The self-trapping effect of tail states is considered as the root of failure[6]. Actually, amorphous materials have been demonstrated to possess a mass of active sites and achieve outstanding grades in many electrochemical processes, even superior to their crystalline counterparts. For example, Idota et al.[7] employed amorphous Sn-based oxide as high-capacity Li-ion-storage material. Li et al.[8] fabricated supercapacitors with ultrahigh energy density using amorphous $Ni(OH)_2$ as the electrode material. Smith et al.[9] synthesized various amorphous metal oxides for electrocatalytic water splitting. Yan et al.[10] used amorphous NiO as a $H_2$ evolution electrocatalyst, and ascribed the active sites to the amorphization-induced under-coordinated Ni atoms ($Ni^0$-like defects). It was reported that such $Ni^0$-like defects would result in electron doping in NiO, which may compensate the intrinsic hole doping[11]. Further considering the correlation between photocatalysis and electrochemistry[12], to exploit possibility of amorphous NiO as an advanced photocatalyst is thus desired[13,14].

To overcome the Achilles' heel of amorphous NiO upon photocatalysis above, here, we propose a strategy taking advantage of the two-dimensional (2D) effect to activate 2D amorphous NiO nanoflakes (2DA) for photocatalysis. We desmonstrate that the 2D effect not only strengthens the electron doping to compensate the intrinsic hole doping of NiO, but also suppresses the tailing-induced carrier recombination of amorphous materials. As a consequence, 2DA can act as an individual reactor without any cocatalysts for solar $H_2$ evolution. Further, surface plasmon resonance (SPR) is introduced by increasing the electron doping, and 2DA is transformed into 2D plasmonic amorphous NiO (2DPA), resulting in the boosting of solar $H_2$ evoultion through the antenna effect of SPR-mediated charge releasing. Note that all previous plasmonic photocatalysts depend on the antenna-reactor heterostructure, where a plasmonic material acts as the antenna to harvest light and another semiconductor or metal acts as the reactor to provide active sites, e.g. Al/Pd, Bi/BiO, $Cu/TiO_2$, Au/CdSe, $W_{18}O_{49}/g-C_3N_4$ heterostructures[15–19]. It is more practical to construct an incorporate antenna-reactor plasmonic structure for solar $H_2$ evolution. Overall, these findings realize the goal of activating amorphous NiO for solar $H_2$ evolution, and provide ideas for the design of incorporate antenna-reactor plamonic photocatalysts.

## Results
**Preparation of amorphous NiO nanoflakes**. 2DA and 2DPA were respectively prepared by laser ablating bulk crystalline NiO powders (labelled as BC) in water and methanol solution[20]. Transmission electron microscopy (TEM) images in Fig. 1a, b show that they are both densely wrinkled nanoflakes with a thickness <10 nm. The corresponding selected-area electron-diffraction (SAED) patterns in the insets indicate full halo rings without any distinguishable diffraction spots, revealing the amorphous nature. The disordered atomic arrangement can be observed by high-resolution TEM (HRTEM) images in Fig. 1c, d.

Obscure diffraction rings in the Fast Fourier transform (FFT) pattern (the inset) demonstrate the existence of small ordered nanodomains in 2DA. Meanwhile, only a full halo ring can be found for 2DPA, suggesting that 2DPA is more disordered. It is further demonstrated by the X-ray diffraction (XRD) patterns in Supplementary Fig. 1, where 2DPA possesses a broadened and right-shifted bread peak due to the decrease of lattice parameter[21]. For comparison, BC and 2D crystalline NiO (2DC) were investigated (Supplementary Note 1). BC is irregular with a size of several hundred nanometers while 2DC shows a morphology like 2DA (Supplementary Fig. 2a–f). Brunauer–Emmett–Teller (BET) analyses manifest that the specific surface areas are respectively 175.93, 163.46, 148.52 and 8.27 $m^2$/g for 2DPA, 2DA, 2DC, and BC.

Raman spectra in Fig. 1e demonstrate the disordered structures further, taking those of BC and 2DC as a reference (Supplementary Fig. 2g). The broadening of the Raman modes suggests the local lattice imperfections and low crystallinity of 2DA and 2DPA. The vanishing of the two-magnon scattering mode (2 M) illustrates that the ordered domains are rather small since 2 M mode is sensitive to the crystalline size. The significant enhancement of the first-phonon modes (1 P) indicates that 2DA and 2DPA are badly disordered since such disorder-induced 1 P modes are not expected in perfect cubic NiO structures[22]. As shown in the Fourier transform infrared (FTIR) spectra in Fig. 1f, the band near 450 $cm^{-1}$ is attributed to the stretching vibration mode of Ni–O bond, which is sensitive to the crystalline order. Compared with BC and 2DC (Supplementary Fig. 2h), 2DA and 2DPA possess broadened Ni–O bands on account of the amorphous nature. The other dinstinct band at 1610 $cm^{-1}$ for 2DPA should be ascribed as the Ni–H bonds[23].

H doping in 2DPA can be verified by the solid-state proton nuclear magnetic resonance ($^1H$ NMR) spectra in Fig. 1g. Clearly, all of the peaks are broad, which should be due to the amorphous nature of 2DA and 2DPA and the resulting various chemical environments for H atoms[24]. Three common peaks can be found for 2DA and 2DPA (labelled as A, B and C). They can be respectively assigned to the terminal Ni–OH, internal Ni–OH and surface adsorbed –OH groups[25]. Since surface Ni atoms of NiO nanoflakes are generally under-coordinated, terminal Ni–OH groups form more easily and peak A is much larger than peak B. In contrast to 2DA, 2DPA exhibits two additional resonances at chemical shifts of 3.4 and −8.4 ppm. They should be associated with the bound H atoms on Ni metal atoms[26]. In detail, they are respectively terminal and internal Ni–H groups[27,28]. Internal Ni–H groups are much more than terminal ones, suggesting that H atoms have been doped into the NiO lattices. At this point, the presence of Ni–H bonds in 2DPA has been demonstrated.

Considering the different atomic structures of the samples, XPS measurements were conducted (Supplementary Fig. 3). Except for C element from the surroundings and Si element from the substrate, only Ni and O elements can be found in the XPS spectra, suggesting that the surfaces of the samples are clean. According to the high-resolution Ni $2p_{3/2}$ XPS spectra in Fig. 1h and Supplementary Fig. 4, the deconvolution results are shown in Supplementary Table 1 (detailed discussion can be found in Supplementary Note 2). Both BC and 2DC possess $Ni^{2+}$ vacancy-induced $Ni^{3+}$ ions since NiO is intrinsically a nonstoichiometric semiconductor[29,30]. It is also the reason for the hole doping nature of crystalline NiO. Interestingly, $Ni^0$-like defects appear in 2DC, 2DA and 2DPA, and the content is increased successively.

O 1 s XPS spectra were analyzed to investigate the origin of $Ni^0$-like defects (Fig. 1i, Supplementary Fig. 4). Five O species can be found, involving under-coordinated nickel oxide ($NiO_{1-x}$), stoichiometric nickel oxide (NiO), over-coordinated nickel oxide ($NiO_{1+x}$), nickel-hydroxyl group (Ni–OH) on the surface and

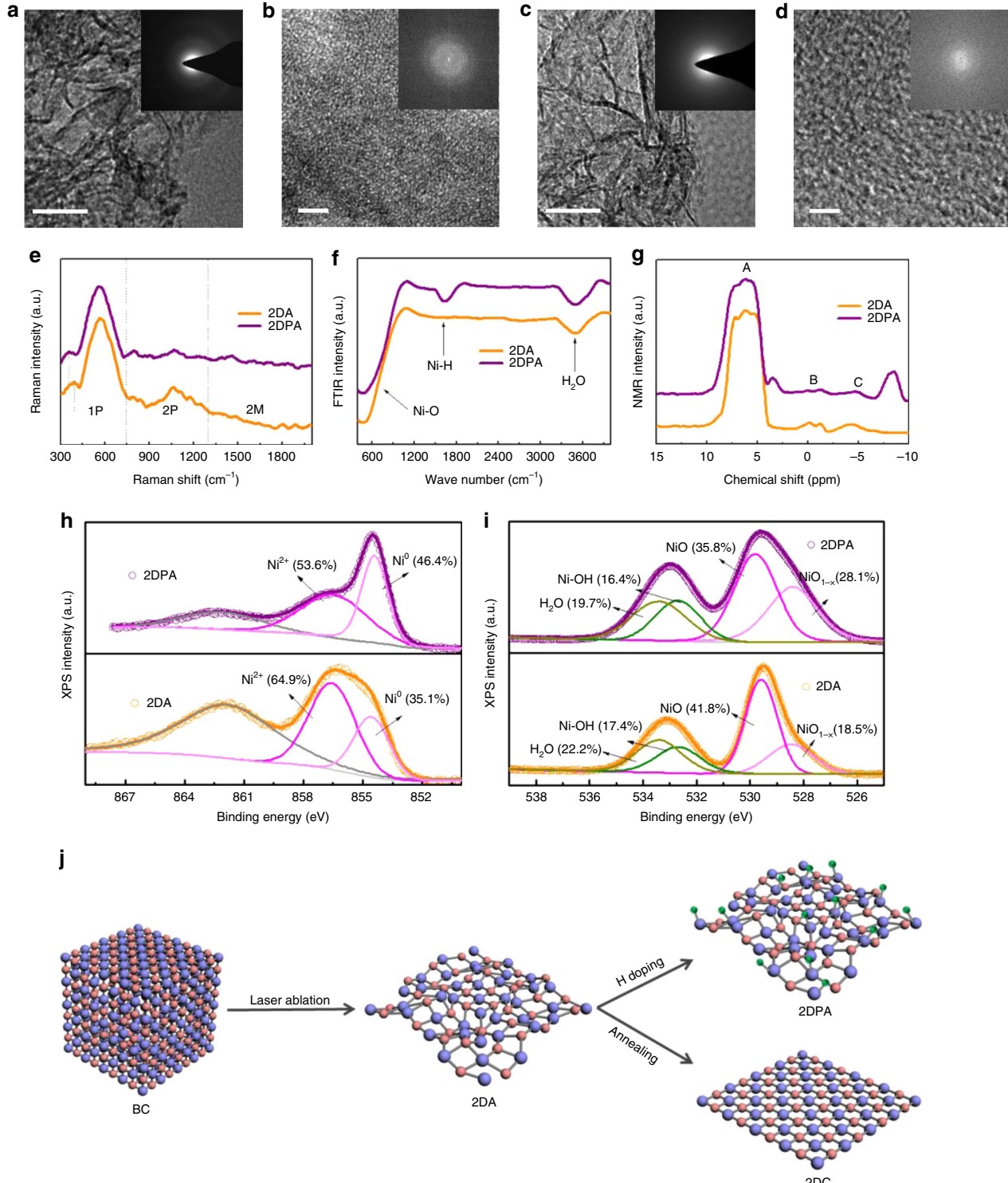

**Fig. 1** Morphologies and structures of 2DA and 2DPA. **a**, **b** TEM images (insets: SAED patterns) of 2DA (**a**) and 2DPA (**b**). **c**, **d** HRTEM images (insets: FFT patterns) of 2DA (**c**) and 2DPA (**d**). Scale bars, 20, 2, 20 and 2 nm in **a**, **b**, **c** and **d**, respectively. **e–i** Raman, FTIR, $^1$H NMR, Ni 2p$_{3/2}$ XPS spectra and O 1 s XPS spectra of 2DA (orange) and 2DPA (purple). **j** Schematic representation of the preparation processes of the four samples. Ni: blue balls, O: pink balls, H: cyan balls

adsorbed $H_2O$. The deconvolution results are shown in Supplementary Table 2. It is known that the edge atoms are generally under-coordinated in the surface shell of a material. For NiO, the under-coordinated Ni atoms (Ni$^0$-like defects) may exist in two forms: NiO$_{1-x}$ and/or Ni–OH. Compared with BC, 2DC possesses an open-framework construction endowing abundant

edge atoms[31]. The Ni–OH and NiO$_{1-x}$ contents in 2DC are increased due to the 2D effect. Considering that the morphologies of 2DA and 2DC are similar, the increase of the Ni$^0$-like defects in 2DA should be due to the amorphization-induced atom rearrangement and coordination defects, which has been reported before[10,32,33]. In 2DPA, due to the electron-donating of the doped

H atoms, the connected $Ni^{2+}$ is reduced and under-coordinated with O atoms. New $Ni^0$-like defects thus form[34]. One can find that the $Ni^0$-like defects are further increased in 2DPA along with the great increase of $NiO_{1-x}$ and slight decrease of Ni-OH[35].

The preparation process of the four samples is shown in Fig. 1j. 2DA and 2DPA are respectively prepared by laser ablating BC in water and methanol-water mixture (Supplementary Fig. 5, Supplementary Note 3). Since the quenching time during the intermission of laser pulses is extremely short (about tens of picoseconds), the random distribution of Ni and O atoms in the produced plasma plumes is frozen, making the products amorphous. It is very similar to the typical cook and quench preparation method for amorphous materials. Finally, -OH and $H_2O$ are adsorbed onto the surface to compensate the surface dangling bonds. As the smallest atoms in nature, H atoms in the liquid can migrate into the plasma plumes. They can be adsorbed on the Ni atoms due to the strong interaction between Ni and H atoms[2,3]. As a result of the reducing environment created by methanol, the connected $Ni^{2+}$ is reduced and Ni-H bonds form in 2DPA. Similar H doping phenomenon has been observed by Hameed et al.[36]. Besides, 2DC is prepared by annealing 2DA in $N_2$. Annealing leads to the rearrangement of Ni and O atoms in 2DA to the stable state. Therefore, crystallization takes place. The coordination defects are partially repaired, resulting in the decrease of $Ni^0$-like defects in 2DC.

**Electronic structures based on spectral characterizations**. The flat-band potentials of the four samples are determined by the Mott–Schottky plots in Fig. 2a, b and Supplementary Fig. 6. The plots of BC, 2DC and 2DA show a negative slope, which is typical for $p$-type semiconductors. On the contrary, 2DPA owns a plot with a positive slope, indicating its $n$-type characteristic. The

Fermi levels are respectively estimated as $+1.67$, $+1.21$, $+0.85$ and $-1.17$ V (vs. NHE) according to the flat-band potentials. Carrier density $N_c$ can be calculated from the plot slope using the equation:[37]

$$N_c = 2e_0^{-1}\varepsilon^{-1}\varepsilon_0^{-1}\left|d\left(C^{-2}\right)/dV\right|^{-1} \qquad (1)$$

where $e_0$ is the electron charge, $\varepsilon_0$ is the vacuum permittivity, $\varepsilon = 25$ is the dielectric constant of NiO[38]. Therefore, hole densities of BC, 2DC and 2DA are respectively $3.7\times10^{18}$, $8.9\times10^{17}$ and $1.9\times10^{17}$ $cm^{-3}$, demonstrating that the intrinsic hole doping of NiO is compensated by electron doping. Electron density of 2DPA is calculated as $8.0\times10^{21}$ $cm^{-3}$. It suggests that the hole doping is entirely compensated and excess free electrons are existing in 2DPA.

It is believed that a carrier density over $10^{21}$ $cm^{-3}$ generally leads to the collective oscillations of free carriers, called as surface plasmon resonance (SPR). As shown in the absorbance spectrum in Fig. 2c, an obvious peak can be found at 529 nm for 2DPA. According to the Drude model[39], the bulk and surface plasmon frequencies ($\omega_p$ and $\omega_{sp}$) can be determined by

$$\omega_p = \sqrt{N_c e^2/\varepsilon_0 m_e} \qquad (2)$$

$$\omega_{sp} = \omega_p/\sqrt{1+\varepsilon_m} \qquad (3)$$

where $m_e$ is the effective mass of an electron, $\varepsilon_m$ is the medium permittivity (in air, $\varepsilon_m = 1$)[40]. The SPR energy is thus calculated as 2.35 eV (528 nm), close to the experimental value (529 nm). The SPR effect is further demonstrated by the surface enhanced Raman scattering (SERS) in Supplementary Fig. 7. Due to the low concentration of the 4-methylbenzenethiol (4-MBT) analyte,

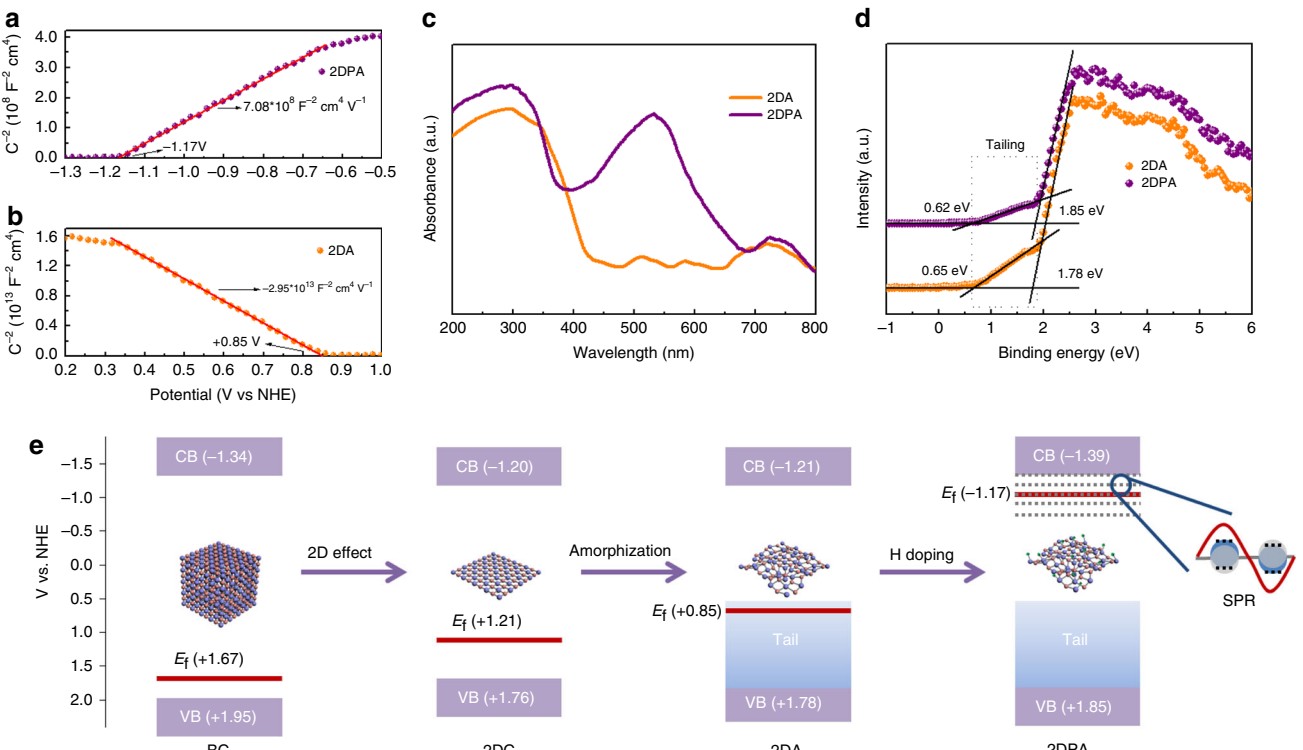

**Fig. 2** Electronic structures based on spectral characterizations. **a**, **b** Mott-Schottky plots of 2DPA (**a**) and 2DA (**b**). $V$ is the applied bias at the electrode, and $C$ is the obtained capacitance. **c**, **d** Absorbance spectra, VB XPS spectra of 2DPA (purple) and 2DA (orange). **e** Schematic illustration of the band structures. The cyan lines represent Fermi levels, and the numbers above mean the Fermi level positions. The light purple bands represent CBs and VBs, and the numbers inside mean the values of CBMs and VBMs

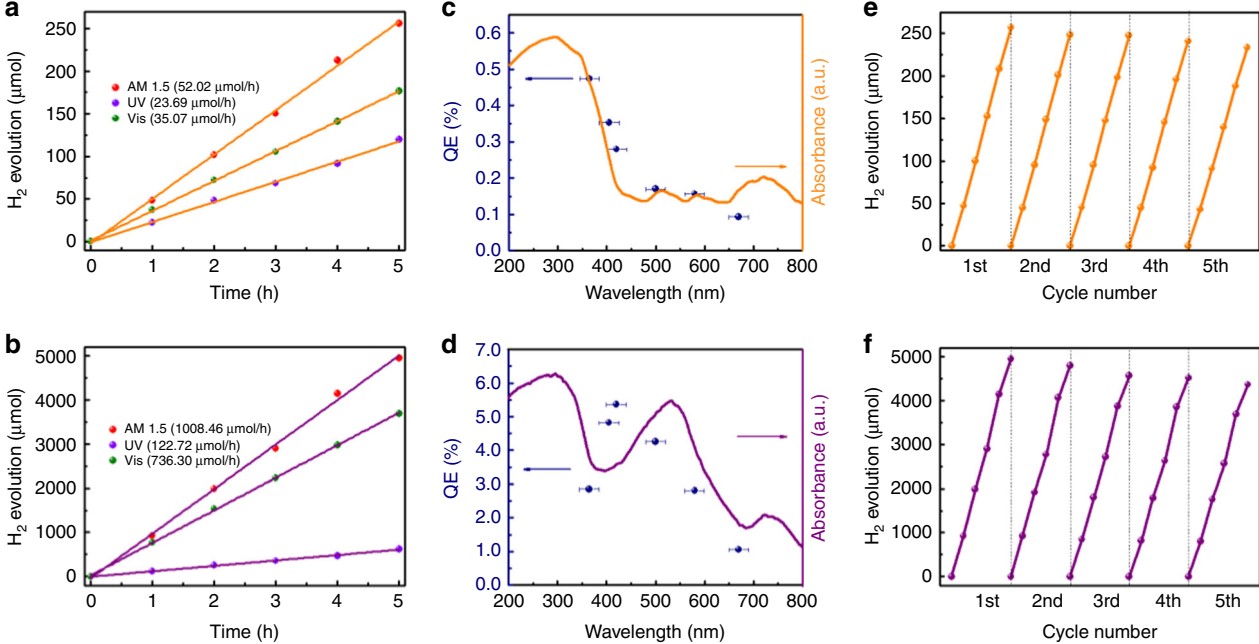

**Fig. 3** Photocatalytic $H_2$ evolution activities evaluation. **a**, **b** Typical time courses of $H_2$ evolution upon AM 1.5, UV and Vis irradiations over 2DA (**a**) and 2DPA (**b**). **c**, **d** Wavelength-dependent QEs (blue dots) of 2DA (**c**) and 2DPA (**d**). The corresponding optical absorption spectra are superimposed for comparison. The error bars indicate the bandwidths of the filters used. **e**, **f** Cycling tests of photocatalytic $H_2$ evolution upon AM 1.5 irradiation over 2DA (**e**) and 2DPA (**f**)

Raman is not visible in the absence of 2DPA. On the contrary, four apparent Raman modes are obtained in the presence of 2DPA, which can be ascribed as the characteristic peaks of 4-MBT[41]. The SERS effect should be because of the local electromagnetic field enhancement of the plasmonic 2DPA[42].

Looking back to the absorbance spectra (Fig. 2c, Supplementary Fig. 8), one can find obvious interband absorption edges. They appear at about 400, 460, 450 and 410 nm repectively for BC, 2DC, 2DA and 2DPA. The optical band gaps are determined from the Tauc plots in Supplementary Fig. 9 to be respectively 3.29, 2.96 and 2.99 and 3.24 eV (Supplementary Note 4). The absorption at around 720 nm should originate from the intra-3$d$ transition in $Ni^{2+}$ incomplete $d$-shell[38]. The additional peaks of 2DA ranging from 450 to 600 nm result from the tail states, which is common in amorphous materials. Tailing effect is further demonstrated by the valence band (VB) XPS spectra[13], shown in Fig. 2d. Prolonged band tails ending at 0.65 and 0.62 eV can be observed for 2DA and 2DPA, respectively. In contrast, no tails can be found for BC and 2DC (Supplementary Fig. 10). The VB values of the four samples are determined as 1.95, 1.76, 1.78 and 1.85 V (vs. NHE), respectively. The conduction band (CB) positions are further estimated from the VB and bandgap values. Therefore, the band structures are schematically illustrated in Fig. 2e. Electron doping induced by 2D effect compensates the intrinsical hole doping of BC and elevates the Fermi level. Due to the abundant surface dangling bonds of 2DC, the existing bonds of the under-coordinated atoms tend to contract to compensate the dangling bonds. This lattice strain will result in the broadening of both VB and CB, pushing the CB minimum (CBM) and VB maximum (VBM) towards the center of the gap and reducing the band gap[31]. Amorphization strengthens electron doping further and generates tail states near VB in 2DA. The bandgap is slightly widened because of the generation of tail states and the pushing of the mobility edges away from the center of the gap. H doping compensates hole doping entirely and produces a mass of free electrons, endowing 2DPA with a strong

SPR at 529 nm. Note that SPR has rarely been realized in amorphous materials before[43,44].

**Photocatalytic activity evaluation.** Photocatalyst powders were dispersed in water containing methanol as a hole scavenger. The $H_2$ evolution activities were evaluated upon simulated solar light (AM 1.5) irradiation. Unsurprisingly, neither BC nor 2DC shows photocatalytic activity. As shown in Fig. 3a, b, both 2DA and 2DPA display continuous $H_2$ evolution. The photocatalytic rate of 2DPA is 1008.46 $\mu$mol h$^{-1}$ 19.4 times larger than that of 2DA (52.02 $\mu$mol h$^{-1}$). For comparison, the photocatalytic reactions were performed in the dark, and no $H_2$ was released, indicating that the $H_2$ evolution is a photo-driven process. The blank reaction without any catalysts was carried out and it showed that no $H_2$ was produced, indicating that the $H_2$ evolution is catalyzed by photocatalysts. Extracting ultra-violet (UV) light from AM 1.5 to trigger the reactions, the $H_2$ evolution rates of 2DA and 2DPA become 23.69 and 122.72 $\mu$mol h$^{-1}$, respectively. Upon visible (Vis) light irradiation individually, the rates are respectively 35.07 and 736.30 $\mu$mol h$^{-1}$. Near-infrared (NIR) light cannot trigger the reactions. Figure 3c, d presents the quantum efficiencies (QEs) at different monochromatic light. For 2DA, QE decreases with increasing wavelength. The wavelength-dependent QE variation trend is nearly consistent with the absorbance spectrum. However, this phenomenon cannot be observed in 2DPA, suggesting the different photocatalytic mechanisms. Further, the activities of 2DA and 2DPA after five cycles show little deterioration (Fig. 3e, f), demonstrating their photo-stability.

As we know, the origin of the evolved $H_2$ is somewhat unclear until now when alcohols are used as hole scavengers. Researchers wonder whether the $H_2$ evolution is a water splitting reaction or a photocatalytic reforming process (Supplementary Note 5)[45]. In this work, no $H_2$ was released when conducting the experiments in pure methanol, suggesting the above photocatalytic process is not a methanol decomposition reaction[46,47]. It is reported that $H_2$ can

also be produced through the methanol reforming reaction

$$CH_3OH + H_2O \xrightarrow{h\nu, catalyst} CO_2 + 3H_2 \qquad (4)$$

Therefore, $CO_2$ was detected, as well. As shown in Supplementary Fig. 11, $CO_2$ was released over 2DA and 2DPA during photocatalytic $H_2$ evolution. The $CO_2$ production rates are 7.39 and 132.31 $\mu mol\,h^{-1}$ respectively for 2DA and 2DPA. The corresponding ratios of evolved $H_2$ to $CO_2$ are 7.04 and 7.62. The value is much larger than that in the proposed methanol reforming reaction (3.0).

Typically, to achieve an optimal performance in the methanol reforming reaction, a large methanol concentration, even larger than 95 vol%, is required since the main species adsorbed on the photocatalyst should be methanol[45,48–50]. Supplementary Fig. 12 exhibits the effect of the methanol concentration on the $H_2$ evolution. Both 2DA and 2DPA obtain the optimal $H_2$ evolution rate when the methanol concentration is low (20%). Excess methanol will compete with water on the adsorption on the photocatalysts, resulting in a lower rate. It suggests that the photocatalytic behavior is not just a methanol reforming process. H atoms in $H_2O$ should make a great contribution on the produced $H_2$. To eliminate the influence of H atoms in methanol, H-free hole scavenger ($Na_2S/Na_2SO_3$) was employed[51]. As shown in Supplementary Fig. 13, both 2DA and 2DPA still show favorable photocatalytic activities with $H_2$ evolution rates of 48.90 and 852.54 $\mu mol\,h^{-1}$. These results reveal that 2DA and 2DPA are efficient photocatalysts to realize $H_2$ evolution from water.

Photocatalytic $O_2$ evolution, the other half-reaction of overall water splitting, was performed in $AgNO_3$ aqueous solution. Unfortunately, no $O_2$ signal was detected. TEM images of the two samples after the $AgNO_3$ sacrificial reactions, labelled as 2DA-A and 2DPA-A, are shown in Supplementary Fig. 14a, b. Obviously, Ag nanoparticles (black dots) have been deposited onto the surfaces of the photocatalysts, which is further demonstrated by the XRD patterns in Supplementary Fig. 15. The peaks at 38.4, 44.1 and 66.3° can be ascribed to the cubic Ag phase (JCPDS 04-0783). 2DPA-A possesses more Ag nanoparticles due to its stronger light harvesting ability. Moreover, methanol was added into the $AgNO_3$ solution to consume the holes, and the samples after the reactions are labelled as 2DA-B and 2DPA-B, respectively. It can be clearly found that more Ag nanoparticles are deposited (Supplementary Fig. 14c, d). The results confirm the electron transfer behavior upon irradiation, which could be enhanced by consuming the holes.

In spite of the $H_2$ evolution activity, photocatalytic $O_2$ evolution was not observed. The fate of the photo-generated holes was thus investigated by monitoring the reactive oxygen species (ROS). Evidently, $\cdot O_2^-$ radicals, $H_2O_2$ and OH radicals were detected in the $AgNO_3$ solution containing 2DA or 2DPA upon irradiation, whereas $^1O_2$ radicals were absent (Supplementary Fig. 16, Supplementary Note 6). In view of the band structures and involved redox potentials (Supplementary Fig. 17), a possible ROS evolution route ($H_2O \rightarrow \cdot O_2^- \rightarrow H_2O_2 \rightarrow \cdot OH$) is proposed (for details, see Supplementary Fig. 18 and Supplementary Note 7)[52,53]. Among these ROS, $H_2O_2$ is the most stable. Therefore, the whole process can be described simply by the equation

$$2Ag^+ + 2H_2O \rightarrow 2Ag + H_2O_2 + 2H^+ \qquad (5)$$

According to the calibration curve in Supplementary Fig. 19, the $H_2O_2$ concentration are respectively determined as 47.07 and

260.08 $\mu mol\,L^{-1}$ for 2DA-A and 2DPA-A. The $H_2O_2$ amounts are respectively 4.71 and 26.01 $\mu mol$. Inductively coupled plasma-mass spectrometry (ICP-MS) was employed to analyze the Ag amounts. They are respectively 1.52 mg (14.07 $\mu mol$) and 6.41 mg (59.35 $\mu mol$), which are approximately twice the $H_2O_2$ amounts. These results demonstrate the proposed reaction route.

Photocatalytic overall water splitting over 2DA and 2DPA was evaluated, as well. Considering the fact that $H_2$ and $O_2$ cannot be produced in pure water, the ROS were also monitored. As shown in Supplementary Fig. 20, $O_2^-$ and OH radicals can still be found. However, $H_2O_2$ disappeared. We consider that the ROS evolution route in pure water may be similar to that in $AgNO_3$ solution ($H_2O \rightarrow \cdot O_2^- \rightarrow H_2O_2 \rightarrow \cdot OH$). The absence of $H_2O_2$ may be due to the decomposition of $H_2O_2$ into OH. Excess OH will be reduced by electrons through the following reaction

$$H^+ + \cdot OH + e^- \rightarrow H_2O \qquad (6)$$

Therefore, $O_2$ was not produced, and $H_2$ evolution was hindered. Proton reduction and water oxidation cannot take place simultaneously (for details, see Supplementary Fig. 21 and Supplementary Note 8).

**Photocatalytic mechanism exploration.** It is well-known that photocatalysis is a surface reaction, i.e., only the photo-generated carriers at the solid–liquid interface can drive the redox reactions[54]. Surface photovoltage (SPV) spectra were employed to probe the surface charges[55]. As shown in Fig. 4a, both UV and Vis irradiations could excite SPV signals. The wavelength-dependent SPV spectra track the corresponding absorbance spectra. BC and 2DC show weak signal intensities, indicating the strong carrier recombination. On the contrary, remarkable responses are obtained by 2DA and 2DPA, which is beneficial to following proton reduction. Interestingly, 2DPA displays a large Vis irradiation-induced photovoltage when SPR is excited, even larger than that of interband transition upon UV irradiation. It suggests that the SPR electrons are not consumed quickly, which has rarely been observed in metal plasmonic materials[56]. Carrier recombination is generally considered as the bottleneck limiting the photon conversion efficiency in photocatalysis, especially for amorphous materials on account of the self-trapping effect of tail states. Photoluminescence (PL) spectroscopy was used to study the radiative recombination process as shown Fig. 4b. Under the excitation at 325 nm, BC and 2DC present strong PL whereas no PL signals can be detected in 2DA and 2DPA. The quenching of PL manifests the suppression of radiative carrier recombination in 2DA and 2DPA.

The active sites on the samples can be assessed by the electrocatalytic $H_2$ evolution performance[57]. Figure 4c shows the electrochemical polarization curves of BC, 2DC, 2DA and 2DPA electrodes. Obviously, they exhibit activity in electrocatalytic $H_2$ evolution. The overpotentials at a current density of 10 $mA\,cm^{-2}$ are respectively 455, 354, 187 and 130 mV. The derived Tafel plots are shown in Supplementary Fig. 22. By fitting the linear portions, the Tafel slopes for the four samples are determined to be 96.3, 77.5, 42.0 and 31.9 $mV\,dec^{-1}$, respectively. The exchange current densities are respectively $1.91 \times 10^{-4}$, $2.66 \times 10^{-4}$, $3.43 \times 10^{-4}$ and $8.53 \times 10^{-4}\,mA\,cm^{-2}$, which are determined by extrapolating the Tafel plots. Considering the linear correlation between the amount of active sites and the exchange current density, the improved electrocatalytic performance can be ascribed to the increased active sites[8,10]. It should be related to the $Ni^0$-like defects since under-coordinated Ni atoms usually play the role of active sites in many Ni-based electrocatalysts[58].

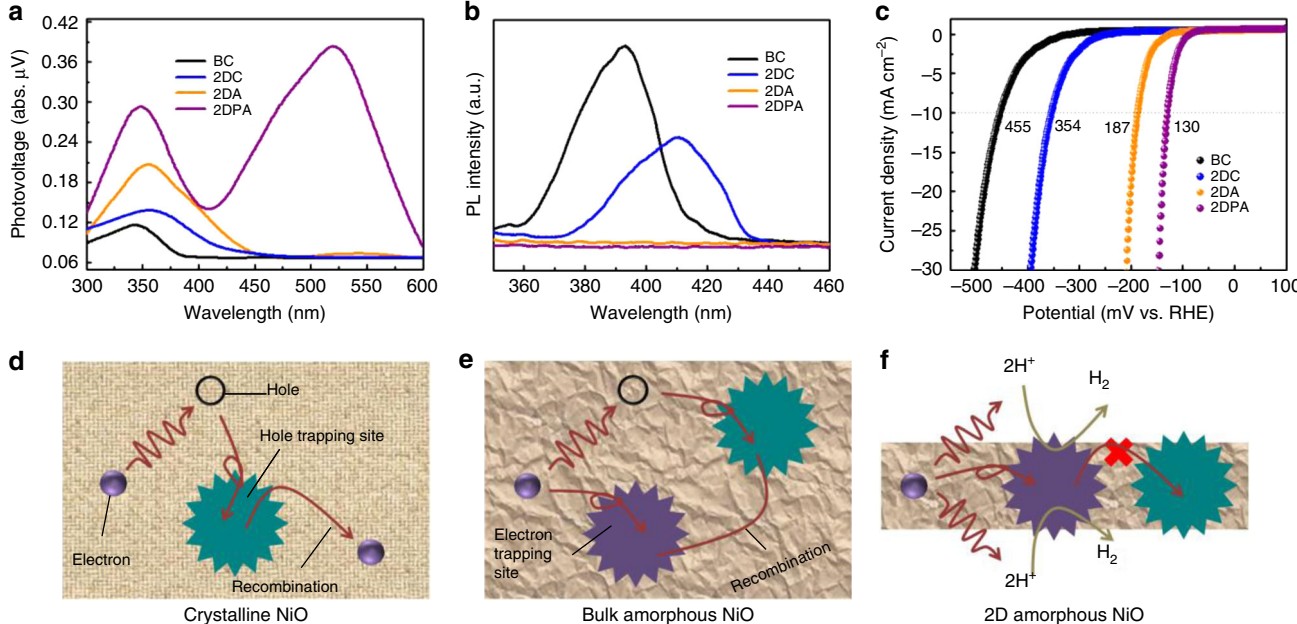

**Fig. 4** Exploration of the photocatalytic mechanism. **a** SPV spectra, **b** PL spectra and **c** electrocatalytic polarization curves of the BC (black), 2DC (blue), 2DA (orange) and 2DPA (purple). **d**–**f** Schematic illustration of the photocatalytic H$_2$ evolution mechanism over crystalline NiO (**d**), bulk amorphous NiO (**e**) and 2D amorphous NiO (**f**)

It is reported that VB of crystalline NiO is dominated by the O 2p states mixed with the Ni 3d states. NiO is intrinsically a nonstoichiometric semiconductor with the Ni$_{0.999}$O composition[30]. The Ni defects lead to the formation of strongly-trapped holes in the VB with an extremely small hole diffusivity[54] of ~$10^{-17}$ m$^2$ s$^{-1}$. Photo-generated carriers will recombine quickly in crystalline NiO, leading to the photocatalytic inertness of BC and 2DC (Fig. 4d). In this case, 2DA and 2DPA are efficient photocatalysts. It seems to be in contradiction with the common concept that amorphous materials are photocatalytically-inactive. Here, we establish that the tailing-induced deactivation phenomenon is prominent in bulk amorphous materials (Fig. 4e). It would be different when compressing one of the dimensions to the scale of a few nanometers (Fig. 4f).

Upon AM 1.5 irradiation, both extended (band states) and localized carriers (tail states) are activated. The diffusion length $L_{ex}$ for an extended electron is on the order of ~ 10 nm, which is estimated by

$$L_{ex} = \sqrt{D_{ex}\tau} \tag{7}$$

where $D_{ex}$ and $\tau$ are the extended electron diffusivity ($10^{-7}$ m$^2$ s$^{-1}$) and time scale ($10^{-9}$ s) for carrier recombination[55,59]. For bulk amorphous materials, extended electrons in the bulk are inclined to recombine with holes before reaching the surface. On the contrary, extended electrons can transport to the surface effectively in 2D amorphous nanoflakes with a thickness less than 10 nm. The 2D channels reduce the carrier recombination probability in significant measure.

As for localized electrons, they are confined on the trapping sites. Thermodynamically, there is an energy barrier between the trapped electrons and holes stopping the recombination. Kinetically, the self-trapping effect decelerates the carrier transport from the trapping sites to the surface. They are still likely to encounter with holes in spite of the energy barrier. The recombination probability is still very large in bulk amorphous materials[60]. Actually, the tail state carriers in amorphous materials are quasi-localized with a localized length including several atoms (even over hundreds of atoms)[61]. The localization length is comparable to the thickness of 2DA and 2DPA nanoflakes. Therefore, the localized electrons can be exposed to the surface quickly. Moreover, the energy barrier between the localized states restrains the encounter. The surface trapping sites are transformed from recombination centers to active sites, endowing 2DA and 2DPA with favorable photocatalytic performance.

**SPR-mediated charge releasing**. H doping can lead to the SPR effect in 2DPA and the great improvement of photocatalytic activity. The stability of the Ni-H bonds should be investigated first to demonstrate whether the involved H atoms contribute to the evolved H$_2$. Figure 5a shows a linear correlation between the photocatalytic rate and the incident light intensity, indicating that the H$_2$ evolution over 2DPA is a photo-driven process, and preliminarily excluding the self-corrosion behavior[52]. A long-term photocatalytic test (3 days) was further carried out. As shown in Fig. 5b, there is still a linear correlation in the H$_2$ evolution time course. The average rate can be determined to be 925.24 µmol h$^{-1}$. The slight decrease should be due to the consumption of methanol scavengers. XRD pattern, FTIR and absorbance spectra of 2DPA after the long-term test are shown in Supplementary Fig. 23. Obviously, 2DPA is still amorphous with Ni–H bonds and SPR peak at 529 nm, demonstrating its stability.

It is discussed above that the QE variation trend of 2DA tracks its absorbance spectrum. However, similar phenomenon cannot be found in 2DPA. Surprisingly, Fig. 5c shows that the QE enhancement factor between 2DPA and 2DA can well track the SPR peak in the optical absorption spectrum of 2DPA[18]. It announces the great contribution of SPR effect on the improved photocatalytic activity of 2DPA. It can also be verified by the fact that the photocatalytic rate ratio of 2DPA to 2DA driven by Vis light (21 times) is much larger than that driven by UV light (5

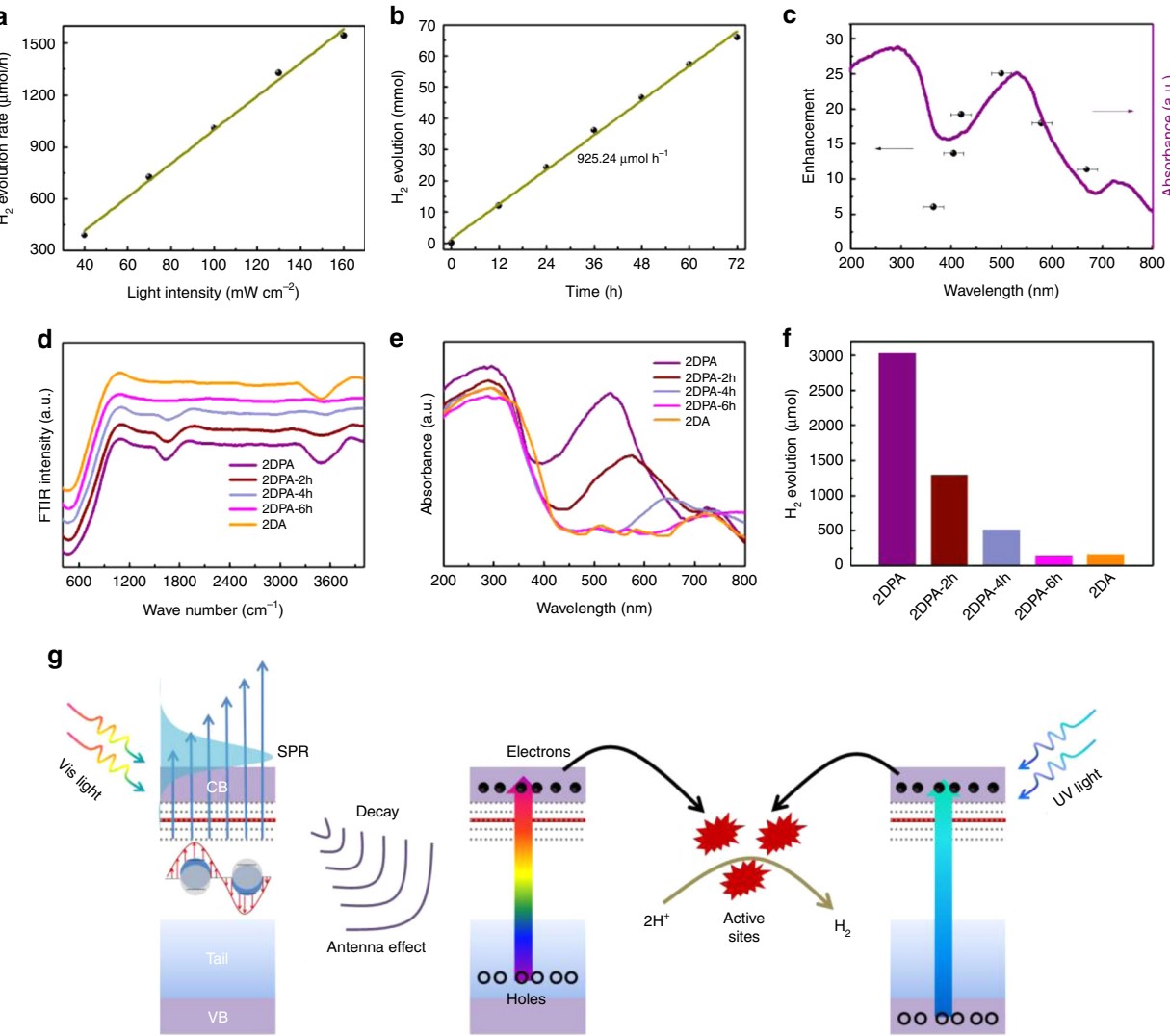

**Fig. 5** SPR-mediated photocatalytic mechanism. **a** Light intensity-dependent $H_2$ evolution rates of 2DPA. **b** Long-term photocatalytic test over 2DPA upon AM 1.5 irradiation. **c** Wavelength-dependent QE enhancement between 2DPA and 2DA (black dots). The error bars indicate the bandwidths of the filters used. **d** FTIR spectra, **e** absorbance spectra and **f** $H_2$ evolution amounts within 3 h of the samples with various H doping amounts. **g** Schematic illustration of the SPR-mediated photocatalytic $H_2$ evolution mechanism

times). Therefore, the SPR-mediated photocatalytic behaviors should be investigated.

2DPA was further annealed at 120 °C in $N_2$ for 2 h, 4 h and 6 h to adjust the H doping amount, and the corresponding samples are labelled as 2DPA-2h, 2DPA-4h and 2DPA-6h. XRD patterns in Supplementary Fig. 24 suggest that they are still amorphous. The intensity of Ni–H bonds in the FTIR spectrum (Fig. 5d) can be the indicator of the H doping amount. One can clearly find that the H doping amount decreases with the prolongation of annealing time. As discussed above, H doping endows 2DPA with a mass of free electrons and a strong SPR at 529 nm. Free electron concentration would be decreased by decreasing the H doping amount. According to the Drude model, the SPR peak would be red-shifted and weakened, which is unsurprisingly reflected in Fig. 5e. Finally, SPR effect vanishes in 2DPA-6h. The photocatalytic $H_2$ evolution amounts within 3 h are summarized in Fig. 5f. Apparently, the weakening of SPR results in the great deterioration of photocatalytic activity. The $H_2$ amount of 2DPA-6h is nearly equal to that of 2DA, suggesting that the leading factor of the photocatalytic behavior over 2DPA is the SPR effect.

A possible photocatalytic mechanism, called as SPR-mediated charge releasing, is proposed (Fig. 5g and Supplementary Note 9). Upon solar irradiation, there are two pathways for harvesting light energy. Interband transition is excited by UV light, and energetic electrons and holes are generated. A mass of free electrons near the Fermi level are excited by vis light (primarily centered at 529 nm), and SPR effect takes place. SPR then decays through Landau damping with the generation of hot electron–hole pairs[40]. It is known that hot carriers can be produced by intraband or interband transitions. Since the Fermi level of 2DPA is much higher than the methanol oxidation potential, the intraband transiton model does not apply in our case[62]. Plasmon decay should take place through interband transition, promoting the releasing of localized tail carriers. Abundant electrons on the CB generated by the two pathways will then transfer to the surface. In this case, 2DPA acts as an antenna to harvest light for carrier generation. 2DPA itself acts as the reactor at the same time to provide active sites for the reactions. As a consequence of the boosting of carrier generation and active sites, the photocatalytic activity of 2DPA is greatly improved.

## Discussion

It should be mentioned that H-doped NiO in our case is different from $Ni(OH)_2$, which has been widely used as a cocatalyst for photocatalytic $H_2$ evolution[48]. Yu et al.[63] has even demonstrated that $Ni(OH)_2$ shows a better cocatalytic activity than NiO. Generally, it would be due to two aspects. For one thing, the light harvesting ability of $Ni(OH)_2$ is rather low, leading to the minimal competitive light absorption of the host photocatalyst[64]. For another, the cocatalytic mechanism of $Ni(OH)_2$ is quite different from that of NiO. NiO acts as a cocatalyst by direct collecting the photo-generated electrons because of its low Fermi level. $Ni(OH)_2$ should be first dissolved. The dissolved $Ni^{2+}$ ions further capture the electrons to produce $Ni^0$ active sites[49]. Unfortunately, these two aspects result in the photocatalytic sluggishness of $Ni(OH)_2$, as well[64]. Therefore, even though $Ni(OH)_2$ performs better than NiO in cocatalysis, further efforts should be made to make it photocatalytically active like NiO[65].

In summary, amorphous NiO is usually evaluated to be unsuitable for photocatalysis due to the intrinsic hole doping of NiO and carrier self-trapping of amorphous nature. However, it can be overcome in 2DA. The intrinsic hole doping is compensated by the electron doping resulting from the amorphization and 2D effect-induced $Ni^0$-like defects. 2D channels suppress carrier recombination, and transform the surface trapping sites from commonly recognized recombination centers to active sites. 2DA is therefore activated for solar $H_2$ evolution. Further, SPR is introduced by increasing the electron doping and 2DA is transformed into 2DPA. An incorporate antenna-reactor structure is therefore constructed to boost the photocatalytic activity through the SPR-mediated charge releasing. Therefore, these findings actually overturn the traditional concept that amorphous materials are photocatalytically-inactive and open up new opportunities for amorphous materials in constructing high-performance photocatalysts.

## Methods

**Amorphous NiO nanoflakes preparation**. 2DA was prepared by laser ablation in liquid (LAL). Typically, crystalline NiO powders (10.0 g, Alfa Aesar, 99.99%) were dispersed in a glass bottle (15 mL) containing 10 mL of deionized water by ultrasonic oscillation (5 min). The bottle was then fixed on a magnetic stirrer with continuous stir. A laser beam was focused into the suspension by a lens with a focus of 10 cm. The focus with a diameter of 3 mm was kept 10 mm below the top of the liquid. The laser beam is the second harmonic from a Q-switched Nd:YAG laser with a wavelength of 532 nm, pulse width of 10 ns, frequency of 10 Hz and single pulse energy of 450 mJ. The LAL process lasted for 6 h. The product was collected by natural precipitation and air drying. For the preparation of 2DPA, 10 mL of deionized water was replaced by 10 mL of 15 vol% methanol aqueous solution.

**Photocatalytic tests**. Photocatalytic $H_2$ evolution was conducted at 4 ℃ in a top irradiation type Pyrex glass vessel connected to a closed gas-circulation system with 30 min degassing pretreatment. 100.0 mg photocatalysts were dispersed in 100 mL aqueous solution containing 20 vol% methanol. A full-spectrum Xenon lamp with an AM 1.5 filter was employed as the light source of simulated solar irradiation (100 mW cm$^{-2}$). Two band-pass filters were used to respectively extract UV and Vis irradiations for photocatalytic reactions. To analyze the wavelength-dependent QEs, various monochromatic filters were used. The amount of photocatalysts was 20.0 mg. All of the reactions were carried out for 5 h. An online gas chromatograph (TCD detector, $N_2$ carrier) was used to detect the gas evolution. The photocatalytic $H_2$ evolution using $Na_2S/Na_2SO_3$ as the hole scavenger was conducted on the same device at 4 ℃. Photocatalyst (100.0 mg) was dispersed in 100 mL $Na_2S/Na_2SO_3$ aqueous solution ($Na_2S$ 0.35 M, $Na_2SO_3$ 0.25 M) upon AM 1.5 irradiation. Photocatalytic $O_2$ evolution was also evaluated on the same device at 4 ℃. The solution was replaced by $AgNO_3$ aqueous solution (0.03 M, 100 mL).

**Reactive oxygen species detection**. The detection of superoxide radicals $O_2{}^-$, singlet oxygen radicals $^1O_2$, hydroxyl radicals OH was performed by the electron spin resonance (ESR)-spin trapping technique on an ESR spectrometer (ER200-SRC, Bruker) using 5-tert-butoxycarbonyl 5-methyl-1-pyrroline N-oxide (BMPO), 2,2,6,6-tetramethyl-4-piperidone (TEMP) and 5,5-dimethyl-1-pyrroline N-oxide (DMPO) as the trapping agents, respectively. Typically, photocatalyst powders

(2DA or 2DPA, 1.0 g L$^{-1}$), $AgNO_3$ (0.03 M) and the corresponding trapping agent (0.05 M) were dispersed in deionized water (100 mL) in the Pyrex glass vessel. After the connection of the vessel to the closed gas-circulation system, degassing pretreatment was conducted for 30 min. After irradiated by AM 1.5 (100 mW cm$^{-2}$) light for 5 min, the solution was quickly sampled into a capillary for the ESR measurements.

The generation of $H_2O_2$ was determined by the coloration method using o-tolidine as the indicator. In detail, 2.0 mL of the suspension was collected immediately after the photocatalytic tests. 0.5 mL of 1% o-tolidine in 0.1 M HCl was then added. After 2 min' standing, the dispersion was acidified with 1 M HCl (2.0 mL). Subsequently, the dispersion was filtered through a membrane filter (0.22 μm). The filtrate was used for the absorbance measurement on a UV–vis spectrophotometer (Lambda9500, PerkinElmer). The concentration of $H_2O_2$ was determined by the calibration curve. $H_2O_2$ solution with different concentration was used as the external standard material. Typically, $H_2O_2$ was added into the 2DPA-containing photocatalytic solution without irradiation. The testing procedure was kept the same.

**Materials characterization**. XRD patterns were obtained on an X-ray diffractometer (D/MAX-2200, Rigaku) at a voltage of 40 V and current of 26 mA with a speed of 10 min$^{-1}$. UV–vis-NIR absorbance spectra of solid powders were recorded on a spectrophotometer (Lambda9500, PerkinElmer) using $BaSO_4$ as a reference. TEM, HRTEM and SAED analyses were carried out on an FTI Tecnai G2 F30 transmission electron microscope. Raman spectra were recorded on a Renishaw InVia Plus laser micro-Raman spectrometer using an argon ion laser (514.5 nm, 90 mW). SERS spectra were recorded on the same system while the power of 514.5 nm laser was tuned to 0.9 mW. Typically, 50 μL 4-methylbenzenethiol (4-MBT) methanolic solution (10 mM) was dried on a blank ITO substrate or an ITO substrate with 1.0 mg 2DPA on it by drop casting. The two sheets were used for SERS measurements. PL measurements were carried out on an InVia Reflex microscope (Renishaw, England) with a He-Cd laser at 325 nm. FTIR spectroscopy was performed in an FTIR spectrometer (Nicolet 6700, Thermo Scientific). XPS and VB XPS spectra were recorded on an XPS scanning microprobe spectrometer (Escalab 250, Thermo-VF Scientific). SPV spectra were obtained with a homebuilt apparatus. BET specific surface areas were determined by the nitrogen adsorption-desorption isotherms on an Automated gas sorption analyzer Autosorb-iQ2-MP at 77 K. Solid-state $^1$H NMR spectra were acquired on a Bruker Ascend-600 spectrometer (600 MHz). ICP-MS measurements were conducted on an ICP-MS device (Thermo Fisher). Mott-Schottky measurements were conducted in a three-electrode electrochemical cell with a Pt counter electrode and saturated Ag/AgCl reference electrode in the dark under $N_2$ bubbling. The electrolyte was 1 M NaOH aqueous solution (pH = 13.6). The frequency was 10 kHz. Electrocatalytic $H_2$ evolution was carried out in the same three-electrode system with Pt sheet as the counter electrode and an Ag/AgCl rod as the reference electrode at room temperature in 1.0 M KOH solution with a scan rate of 5 mV s$^{-1}$ and iR corrections. The sample powder (10.0 mg) was dispersed into a mixed solution with 450 μL ethanol and 50 μL Nafion. The working electrode was prepared by depositing the as-prepared suspension (30 μL) onto the FTO sheet (1 × 2 cm$^2$).

## Data availability

The data that support the findings of this study are available from the corresponding author upon request.

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

## Acknowledgements

The National Basic Research Program of China (2014CB931700) and State Key Laboratory of Optoelectronic Materials and Technologies supported this work.

## Author contributions

G.W.Y. designed the experiments; Z.Y.L. performed the experiments, calculations and data analysis; C.D., B.Y. and X.C.W. assisted with some of the experiments; Z.Y.L. and G.W.Y. wrote the paper.

## Additional information

**Competing interests:** The authors declare no competing interests.

 **11**