## [Peer Review File · Nature Communications]

Reviewers' comments:

Reviewer #1 (Remarks to the Author):

This manuscript, entitled "Boosting solar H₂ evolution over 2D amorphous NiO photocatalyst by constructing an incorporate antenna-reactor plasmonic structure" addresses the 2D amorphous NiO photocatalyst for the solar H₂ evolution with a detailed characterization and mechanism study, exhibiting very interesting features, which makes it high possibility for publication in Nature Communication. The exhaustive studies on electronic structure, multiple-photoresponse lead to the convincing explanation of the most relevant mechanisms operating in this system. Therefore, I would like to recommend its publication if following concerns can be addressed:

- 1) The formation of Ni-H bonding as well as its formation mechanism are unclear. Since the H doping seems to act as important role for overall photocatalytic performance, author is suggested to give a convincing discussion about it.
- 2) Generally, Ni is one of well-known hydrogen evolution catalysts for H₂ generation. If the electron goes to Ni active site, whether the Ni-H is involved in H₂ generation. Therefore, the Ni-H bonding stability should be checked.
- 3) Until now, it is somewhat unclear, when the methanol is used as hole scavenger, the photocatalytic H₂ is produced from H₂O or methanol, taking the doubts of comment 2 and 3 together, an isotope labelling experiment is suggested to provide.
- 4) Because the 2D amorphous NiO photocatalyst presents photo- and electro-catalytic H₂ behaviours, therefore, the hydrogen evolution activity of NiO photocatalyst should be investigated.

Reviewer #2 (Remarks to the Author):

2D NiO nanosheet was developed as plasmonic assisted photocatalyst. This paper can be accepted after considering follows.

- 1) Already there are many reports on photocatalysis of Ni(OH)₂ nanosheet. What is big difference between present catalyst and these Ni(OH)₂ nanosheet.
- 2) What is evidence of presence of plasmonic effect in the present system. For prove of surface plasmonic effect, enhancement of Raman spectra should be observed.

Reviewer #3 (Remarks to the Author):

This paper deals with amorphous NiO_x prepared by laser ablation in the liquids (H₂O, H₂O/CH₃OH). The prepared NiO_x are characterized by various kinds of the spectroscopies (TEM, UV-vis-NIR, Raman, XRD, FTIR, and PL), and their semiconducting, optical and photocatalytic properties are evaluated by the electrochemical methods such as Mott-Schottky and photovoltage measurements. The paper states the findings that the 2D amorphous NiO_x provides high photocatalytic performance with water splitting, in spite of amorphous structures, which is due to the antenna-reactor plasmonic structures. The application of amorphous materials with SPR effects to photocatalysts is intriguing. However, very

complex 2D amorphous structures consisting of various chemical states (Ni, NiO_{1-x}, NiO, Ni-H, NiOH, H₂O) make it difficult to understand the antenna and plasmonic (SPR) roles in photocatalysis, and hence the explanations to the results are considerably ambiguous. The present paper needs experimental-based improvement to reach a scientific level to be accepted.

Main criticisms are as follows.

(1) It is important to characterize 2DPA used for the prolonged photocatalytic reaction and to demonstrate little changes with the photocatalytic reaction. Especially, hydrogen involved in 2DPA and SPR effects should be monitored with the sample after the reaction. The author need to clearly demonstrate that the hydrogen involved in 2DPA has nothing to do with the product (H₂) observed in the photocatalytic water splitting.

(2) It seems difficult to consider the metal-induced plasmonic effects, because Ni existing in 2DA and 2DPA are not grown as metal with a particle size of ~ 1nm or larger (TEM images). The SPR effects of 2DPA is likely to be due to a large amount of H involved, whereas, for 2DA, there are neither H in bulk nor absorption at around 500 nm. The SPR effects for 2DA are questionable.

(3) As one of the feature of the present photocatalysts system, the authors claim the cooperative effects of interband and SPR effects, based on the differences in activity with Mx and Sum (Fig.4 (b)). However, the differences are too small to confirm the effects. For example, at 500nm where Vis has a maximum, both are nearly the same, and at 405 where the largest difference appears, it is only 1.4 fold difference. These results suggest that the on-average differences between Mx and Sum are within experimental error, and the cooperative effects of interband transition and SPR effects are poorly convincing.

(4) How important the amorphous structures are for the generation of photocatalysis in the present system? The 2D flake structures, even if they are crystal, seem enough when the SPR effects exist. The authors should show the important role of amorphous structures in photocatalysis.

(5) Fig. 4(j) shows a change from amorphous (2DA) to crystal (2DC) by annealing. For better understanding, it is more convincing to show how both the photocatalysis and SPR effects change from 2DPA to 2DC (or 2DA) by step-wise gradual annealing (possibly changing annealing temperatures from low temperature up to high temperature), in particular, to see the correlation among hydrogen amount, SPR effects and photocatalysis.

(6) The amorphous NiO_x has no ability for overall water splitting to produce H₂ and O₂, although H₂ is obtained from H₂O/CH₃OH. How does the oxidation reaction proceeds on this photocatalyst? For example, is the evolution of oxygen from Ag(NO₃) aq. solution by UV and Vis light irradiation observed with the wavelength dependence similar to H₂ evolution?

(7) Models of Fig.2(e) and Fig. (c) 4 are very vague. They should be more specific to provide correct information.

We are pleasure to submit the revised manuscript of our original manuscript (title: Boosting solar H₂ evolution over 2D amorphous NiO photocatalyst by constructing an incorporate antenna-reactor plasmonic structure) to *Nature Communications* for publication, and the authors are Zhaoyong Lin, Chun Du, Bo Yan, Chengxin Wang and Guowei Yang. Professor Guowei Yang is the corresponding author.

First of all, we appreciated the comments from the reviewers of our manuscript. Thanks a lot for the reviewers, because we have learned more in science from their review reports. These suggestions greatly help us to improve our work.

According to the reports of the referees, we have done the significant improvement to the original manuscript by adding the new experiments and deeply theoretical analysis into the revised manuscript. We need to point out that a lot of experiments and theoretical analysis have been carefully carried out in the revised manuscript to address all issues from the referees, which guarantees that all the results are credible and replicative in our study. Additionally, we have rewritten many discussions for having a clear and general insight into the fundamental aspects involved in our work based on the suggestions from the referees. Therefore, we believe that the revised manuscript could meet all criticisms from the reviewers, and reach to the scientific standards of *Nature Communications*. Thus, in our opinion, the revised manuscript should be convincing and appropriate for publication by *Nature Communications*.

Considering the contribution of Chengxin Wang to the revised manuscript, all authors have decided to add Chengxin Wang to be one of the authors of the revised manuscript.

Note that the mainly new added paragraphs and rewritten sections were printed in blue color in the revised manuscript.

Please don't hesitate to let me know if you have any comment about our revised manuscript.

Response to Referees

For Reviewer #1

Comments #1. "The formation of Ni-H bonding as well as its formation mechanism are unclear. Since the H doping seems to act as important role for overall photocatalytic performance, author is suggested to give a convincing discussion about it."

Our Response. Thank a lot for this valuable suggestion. Characterizations and discussions about the formation and the formation mechanism of Ni-H bonding have been added in the revised manuscript.

The formation of Ni-H bonds is demonstrated by the FTIR and XPS results. It is generally believed that a water environment is necessary for the formation of amorphous phase by laser ablation on account of its large specific heat capacity for the rapid cooling process. In our case, 2DPA is prepared in methanol-water mixture with a smaller specific heat capacity. Surprisingly, 2DPA is more disordered than 2DA, demonstrated by the TEM, XRD and Raman results. It should be because of the formation of Ni-H bonds (FTIR spectra). It is in accord with our previous work revealing a reducing environment, in this case, methanol, is necessary for the H doping (*Appl. Catal., B* **227**, 35-43 (2018)). As the smallest atoms in nature, H atoms in the liquid can migrate to the lattice of NiO during the quenching step in the laser ablation process. They can be adsorbed on the Ni atoms due to the strong interaction between Ni and H atoms. Adding methanol into the water, a reducing environment is created. As a result of the electron-donating of H and the created reducing environment, the connected Ni²⁺ is reduced and Ni-H bonds form in 2DPA. Similar H doping phenomenon has been observed by Hameed *et al.* (*J. Mol. Catal. A: Chem.* **223**, 35-41 (2005)).

Further, 2DPA was annealed at 120 °C in N₂ for 2h, 4h and 6h to adjust the H doping amount. As shown in the XRD patterns in Supplementary Fig. 13, annealing does not result in crystallization of the 2DPA nanoflakes, and the samples are still amorphous. The intensity of Ni-H bonds in the FTIR spectrum (Fig. 5d) can be the indicator of the H doping amount. One can clearly find that the H doping amount decreases with the prolongation of annealing time. H doping in 2DPA-6h nearly disappears with an intensity of Ni-H bonds identical to that of 2DA. In this work, H doping endows 2DPA with a mass of free electrons and a strong SPR at 529 nm. Free electron

concentration would be decreased by decreasing the H doping amount. According to equations on the SPR peak estimation, the SPR peak would be red-shifted and weakened, which is unsurprisingly reflected in Fig. 5e. Finally, the SPR effect vanishes in 2DPA-6h.

Therefore, the formation and the formation mechanism of Ni-H bonds are demonstrated.

Comments #2. "Generally, Ni is one of well-known hydrogen evolution catalysts for H₂ generation. If the electron goes to Ni active site, whether the Ni-H is involved in H₂ generation. Therefore, the Ni-H bonding stability should be checked."

Our Response. Thanks for this remark. We should acknowledge that the stability of Ni-H bonding is important for the photocatalytic behavior. It is demonstrated that H doping can lead to the SPR effect in 2DPA and the great improvement of the photocatalytic activity. The stability of the Ni-H bonds should be investigated to demonstrate whether the involved H atoms contribute to the evolved H₂.

(a) Fig. 5a shows a linear correlation between the photocatalytic rate and the incident light intensity, indicating that the H₂ evolution over 2DPA is a photo-driven process, and preliminarily excluding the self-corrosion behavior (*Nat. Commun.* **9**, 1397 (2018)).

(b) To directly evaluate the stability of the Ni-H bonds, a long-term photocatalytic test was carried out. As shown in Fig. 5b, there is still a linear correlation in the time course of H₂ evolution for the 3-day long-term test. The average photocatalytic rate can be determined to be 925.24 $\mu\text{mol}/\text{h}$. The slight decrease should be due to the consumption of methanol scavengers. XRD pattern, FTIR spectrum and absorbance spectrum of 2DPA after the long-term test are shown in Supplementary Fig. 12. Obviously, 2DPA is still amorphous with Ni-H bonds and SPR peak at 529 nm, demonstrating its stability.

Comments #3. "Until now, it is somewhat unclear, when the methanol is used as hole scavenger, the photocatalytic H₂ is produced from H₂O or methanol, taking the doubts of comment 2 and 3 together, an isotope labelling experiment is suggested to provide."

Our Response. We appreciated the referee's remark. However, we cannot carry out

the isotope labelling experiment in our lab now due to the technique limited.

As we know, an isotope labeling study is generally employed to determine the origin of the evolved gases in photocatalytic overall water splitting. However, it may be not applicable in our case. As mentioned by Kandiel, H⁺ is easier to be reduced than D⁺ (*Energy Environ. Sci.* **7**, 1420-1425 (2014)). An isotope labeling will influence the reaction pathways to some extent. Further, the rapid and frequent H⁺/D⁺ isotopes exchange will take place in the whole photocatalytic process, making it difficult to determine the origin.

Therefore, we believe that other experiments can also been used to address this issue, i.e., the relevant experiments have been carried out to explore the origin of H atoms in the produced H₂ in the revised manuscript.

In this work, no H₂ was released when conducting the experiments in pure methanol, suggesting the above photocatalytic process is not a methanol decomposition reaction (*Sci. Rep.* **5**, 13475 (2015)). It was reported that H₂ can also be produced through the methanol reforming reaction

It can be divided into four steps:

To achieve an optimal photocatalytic performance, a large methanol concentration, even larger than 95 vol.%, is usually required since the main species adsorbed on the photocatalyst should be methanol (*J. Catal.* **326**, 43-53 (2015)). Supplementary Fig. 10 exhibits the effect of the methanol concentration on the H₂ evolution rates over 2DA and 2DPA. Both 2DA and 2DPA obtain the optimal H₂ evolution rate when the methanol concentration is low (20%). Excess methanol will compete with water on the adsorption on the photocatalysts, resulting in a lower photocatalytic rate. These results thus suggested that the photocatalytic behavior is not just a methanol reforming process.

Accordingly, our experiments have demonstrated that H atoms in H₂O should make a great contribution on the produced H₂ in our case.

Comments #4. "Because the 2D amorphous NiO photocatalyst presents photo- and electro-catalytic H₂ behaviours, therefore, the hydrogen evolution activity of NiO photocatalyst should be investigated."

Our Response. Thanks very much for this suggestion. In our case, we have demonstrated that 2D amorphous NiO nanoflakes (2DA and 2DPA) can act as efficient photocatalysts for H₂ evolution although NiO is usually considered as photocatalytically inactive. In the revised manuscript, the electrocatalytic H₂ evolution activities have also been investigated to assess the active sites on the sample.

Fig. 4c shows the electrochemical polarization curves of BC, 2DC, 2DA and 2DPA electrodes. Obviously, these results exhibit activity in electrocatalytic H₂ evolution. The overpotentials at a current density of 10 mA cm⁻² are respectively 455, 354, 187 and 130 mV. The derived Tafel plots are shown in Supplementary Fig. 11. By fitting the linear portions, the Tafel slopes for the four samples are determined to be 96.3, 77.5, 42.0 and 31.9 mV dec⁻¹, respectively. The exchange current densities are respectively 1.91×10^{-4} , 2.66×10^{-4} , 3.43×10^{-4} and 8.53×10^{-4} mA cm⁻², which are determined by extrapolating the Tafel plots. Considering the linear correlation between the amount of active sites and the exchange current density, the improved electrocatalytic performance can be ascribed to the increased active sites. It should be related to the Ni⁰-like defects since under-coordinated Ni atoms usually play the role of active sites in many Ni-based electrocatalysts (*Appl. Catal., B* **164**, 18-30 (2015)).

For Reviewer #2

Comments #1. "Already there are many reports on photocatalysis of Ni(OH)₂ nanosheet. What is big difference between present catalyst and these Ni(OH)₂ nanosheet."

Our Response. Thanks for this remark. The big difference between NiO catalyst and Ni(OH)₂ nanosheet is that NiO acts as a catalyst in our case and Ni(OH)₂ nanosheet acts as a cocatalyst in previous reports upon photocatalytic H₂ evolution.

As we know, Ni(OH)₂ has been widely used as a cocatalyst for photocatalytic H₂ evolution (*J. Catal.* **326**, 43-53 (2015)), e.g., Yu *et al.* have demonstrated that Ni(OH)₂ shows a better cocatalytic activity than NiO (*Catal. Sci. Technol.* **3**,

1782-1789 (2013)). It would be due to two aspects. One is that the light harvesting ability of Ni(OH)₂ is rather low, leading to the minimal competitive light absorption of the host photocatalyst (*Adv. Mater. Interfaces* **3**, 1600256 (2016)). Another is that the cocatalytic mechanism of Ni(OH)₂ is quite different from that of NiO. NiO acts as a cocatalyst by directly collecting the photo-generated electrons because of its low Fermi level. Ni(OH)₂ should be first dissolved. The dissolved Ni²⁺ ions further capture the electrons to produce Ni⁰ active sites (*ChemistrySelect* **2**, 7497-7507 (2017)).

Comments #2. "What is evidence of presence of plasmonic effect in the present system. For prove of surface plasmonic effect, enhancement of Raman spectra should be observed."

Our Response. Thanks a lot for this suggestion. In fact, in our case, the plasmonic effect has been demonstrated by the Mott-Schottky measurement-based SPR peak estimation.

We think that the surface enhanced Raman scattering (SERS) experiments are helpful (Supplementary Fig. 6) for the understanding of plasmonic effect. Due to the low concentration (10 mM) of the 4-methylbenzenethiol (4-MBT) analyte, the Raman spectrum of 4-MBT is not visible in the absence of 2DPA. On the contrary, four apparent Raman modes are obtained in the presence of 2DPA, which can be ascribed as the characteristic peaks of 4-MBT. The SERS effect should be because of the local electromagnetic field enhancement of the plasmonic 2DPA (*Nanoscale* **6**, 1423-1433 (2014)). Therefore, the SPR effect has been demonstrated directly in the revised manuscript.

For Reviewer #3

Comments #1. "It is important to characterize 2DPA used for the prolonged photocatalytic reaction and to demonstrate little changes with the photocatalytic reaction. Especially, hydrogen involved in 2DPA and SPR effects should be monitored with the sample after the reaction. The author need to clearly demonstrate that the hydrogen involved in 2DPA has nothing to do with the product (H₂) observed in the photocatalytic water splitting."

Our Response. Thanks for this remark. We should acknowledge that the stability of Ni-H bonding is important for the photocatalytic behavior. It is demonstrated that H

doping can lead to the SPR effect in 2DPA and the great improvement of the photocatalytic activity. The stability of the Ni-H bonds should be investigated to demonstrate whether the involved H atoms contribute to the evolved H₂.

(a) Fig. 5a shows a linear correlation between the photocatalytic rate and the incident light intensity, indicating that the H₂ evolution over 2DPA is a photo-driven process, and preliminarily excluding the self-corrosion behavior (*Nat. Commun.* **9**, 1397 (2018)).

(b) To directly evaluate the stability of the Ni-H bonds, a long-term photocatalytic test was carried out. As shown in Fig. 5b, there is still a linear correlation in the time course of H₂ evolution for the 3-day long-term test. The average photocatalytic rate can be determined to be 925.24 $\mu\text{mol} / \text{h}$. The slight decrease should be due to the consumption of methanol scavengers. XRD pattern, FTIR spectrum and absorbance spectrum of 2DPA after the long-term test are shown in Supplementary Fig. 12. Obviously, 2DPA is still amorphous with Ni-H bonds and SPR peak at 529 nm, demonstrating its stability.

Comments #2. "It seems difficult to consider the metal-induced plasmonic effects, because Ni existing in 2DA and 2DPA are not grown as metal with a particle size of $\sim 1\text{nm}$ or larger (TEM images). The SPR effects of 2DPA is likely to be due to a large amount of H involved, whereas, for 2DA, there are neither H in bulk nor absorption at around 500 nm. The SPR effects for 2DA are questionable."

Our Response. Thanks for this remark. In this work, the photocatalytic H₂ evolution has been realized in 2DA. Note that SPR effect can be further introduced by H doping, and 2DA is transformed into 2DPA. It has been demonstrated that SPR effect in 2DPA is due to the large amount of H involved. It is indeed that 2DA does not possess H doping. Therefore, SPR effect cannot be found in 2DA. In the revised manuscript, we have clearly stated that 2DA does not possess SPR effect, and SPR effect can only be found in 2DPA.

Comments #3. "As one of the features of the present photocatalysts system, the authors claim the cooperative effects of interband and SPR effects, based on the differences in activity with Mx and Sum (Fig.4 (b)). However, the differences are too small to confirm the effects. For example, at 500nm where Vis has a maximum, both

are nearly the same, and at 405 where the largest difference appears, it is only 1.4 fold difference. These results suggest that the on-average differences between Mx and Sum are within experimental error, and the cooperative effects of interband transition and SPR effects are poorly convincing."

Our Response. We very appreciated for this remark. It is our mistake to make the previous conclusion about the cooperative effects of interband transition and SPR effects in the original manuscript.

In this revised manuscript, the SPR-mediated photocatalytic mechanism has been re-investigated. A possible photocatalytic mechanism, called as SPR-mediated charge releasing (Fig. 5g), has been proposed. Upon solar irradiation, there are two pathways for harvesting light energy. Interband transition is excited by UV light, and energetic electrons and holes are generated, ready for the photocatalytic reactions. A mass of free electrons near the Fermi level are excited by Vis light (primarily centered at 529 nm), and SPR effect takes place. SPR then decays through Landau damping with the generation of hot electron-hole pairs (*Nat. Commun.* **8**, 15070 (2017)). It is known that hot carriers can be produced by intraband or interband transitions. Since the Fermi level of 2DPA is much higher than the methanol oxidation potential, the intraband transition model may be unsuitable in our case (*J. Mater. Chem. A* **3**, 9491-9501 (2015)). Therefore, plasmon decay should take place through interband transition, promoting the releasing of localized tail carriers. Abundant electrons on the CB generated by the two pathways will then transfer to the surface active sites. In this case, 2DPA acts as an antenna to harvest light for carrier generation. 2DPA itself acts as the reactor at the same time to provide active sites for the photocatalytic reactions. The electrocatalytic behaviors indicate that the amount of active sites of 2DPA is much larger than that of 2DA. As a consequence of the boosting of carrier generation and active sites, the photocatalytic activity of 2DPA is greatly improved.

Comments #4. "How important the amorphous structures are for the generation of photocatalysis in the present system? The 2D flake structures, even if they are crystal, seem enough when the SPR effects exist. The authors should show the important role of amorphous structures in photocatalysis."

Our Response. Thanks for this remark. In the previous reports, NiO crystals fail to act as an individual reactor for photocatalytic H₂ evolution because of the intrinsic hole doping, regardless of their impressive cocatalytic ability for proton/electron

transfer. Amorphous materials are usually evaluated as photocatalytically-inactive due to the amorphous nature-induced self-trapping of tail states, in spite of their achievements in electrochemistry. In this work, we have demonstrated that 2DA can act as an efficient and robust photocatalyst for solar H₂ evolution without any cocatalysts. The amorphization and 2D effect provide abundant Ni⁰-like defects to compensate the intrinsic hole doping. Meanwhile, 2D channels suppress the carrier recombination. Further, the antenna effect of surface plasmon resonance (SPR) can be introduced to construct an incorporate antenna-reactor structure by increasing the electron doping for promoting H₂ production in 2DPA. The solar H₂ evolution rate is improved by a factor of 19.4 through the SPR-mediated charge releasing.

According to the suggestion of the referee, we have tried to realize H doping in 2D crystalline NiO nanoflakes (2DC) by the H-spillover process (*J. Am. Chem. Soc.* **138**, 9316-9324 (2016)). Unfortunately, we failed. H doping still remains a challenging issue for many metal oxides since it usually requires harsh reduction conditions for the H-spillover process.

According to our studies, we can know that the bandgap of crystalline NiO is large (3.29 eV for BC, 2.96 eV for 2DC). Tailing effect cannot be found in them. Therefore, they can only harvest UV light. If SPR effect can be introduced into BC and 2DC, it is hard to realize SPR in the UV region since a mass of free electrons are required. If the SPR peaks for BC and 2DC appear in the visible or NIR regions, they can only damping through intraband transitions rather than interband transitions. At this point, carrier recombination may be severe (*Nanoscale* **8**, 8826-8838 (2016)). Therefore, the photocatalytic activity may be limited.

In this work, we focus on the amorphization and 2D effect can realize photocatalysis in NiO, and further H doping can results in the SPR effect and the improved photocatalytic activity. Further efforts should be made to investigate NiO-based photocatalysts.

Comments #5. "Fig. 4(j) shows a change from amorphous (2DA) to crystal (2DC) by annealing. For better understanding, it is more convincing to show how both the photocatalysis and SPR effects change from 2DPA to 2DC (or 2DA) by step-wise gradual annealing (possibly changing annealing temperatures from low temperature up to high temperature), in particular, to see the correlation among hydrogen amount, SPR effects and photocatalysis."

Our Response. Thank very much for this suggestion. 2DPA was annealed at 120 °C in N₂ for 2h, 4h and 6h to adjust the H doping amount, and the corresponding samples are labeled as 2DPA-2h, 2DPA-4h and 2DPA-6h, respectively. As shown in the XRD patterns in Supplementary Fig. 13, annealing does not result in crystallization of the 2DPA nanoflakes, and the samples are still amorphous. The intensity of Ni-H bonds in the FTIR spectrum (Fig. 5d) can be the indicator of the H doping amount. One can clearly find that the H doping amount decreases with the prolongation of annealing time. H doping in 2DPA-6h nearly disappears with an intensity of Ni-H bonds identical to that of 2DA. Certainly, annealing also leads to the remove of adsorbed H₂O molecules. In this work, H doping endows 2DPA with a mass of free electrons and a strong SPR at 529 nm. Free electron concentration would be decreased by decreasing the H doping amount. According to equations on the SPR peak estimation, the SPR peak would be red-shifted and weakened, which is unsurprisingly reflected in Fig. 5e. Finally, SPR effect vanishes in 2DPA-6h. Further, photocatalytic H₂ evolution was conducted. The H₂ evolution amounts within 3 h are summarized in Fig. 5f. Apparently, the weakening of SPR results in the great deterioration of photocatalytic activity. The H₂ evolution amount of 2DPA-6h is nearly equal to that of 2DA, suggesting that the leading factor of the photocatalytic behavior over 2DPA is SPR effect.

Comments #6. "The amorphous NiOx has no ability for overall water splitting to produce H₂ and O₂, although H₂ is obtained from H₂O/CH₃OH. How does the oxidation reaction proceeds on this photocatalyst? For example, is the evolution of oxygen from Ag(NO₃) aq. solution by UV and Vis light irradiation observed with the wavelength dependence similar to H₂ evolution?"

Our Response. Thanks for this remark. Considering that H₂ and O₂ cannot be produced in pure water without methanol over 2DA and 2DPA, photocatalytic O₂ evolution, the other half-reaction of overall water splitting, was performed in AgNO₃ aqueous solution (0.03 M, 100 mL) using Ag⁺ as the electron scavengers. Unfortunately, no O₂ signal can be detected. NiO has been widely used as H₂ evolution catalysts instead of O₂ evolution catalysts due to its strong interaction with H atoms. The lack of active sites will lead to the failure in O₂ evolution. Moreover, it is known that photocatalytic O₂ evolution is a four-electron process requiring a large overpotential (*Science* **347**, 970-974 (2015)). Fig. 2e indicates that the VBMs of 2DA

and 2DPA are not much lower than the O₂ evolution level (1.23 V vs. NHE). Their O₂ evolution sluggishness may be due to the kinetic limitation of the oxidation process, as well. That would also be the reason for the addition of hole scavengers for photocatalytic H₂ evolution.

Comments #7. "Models of Fig. 2(e) and Fig. (c) 4 are very vague. They should be more specific to provide correct information."

Our Response. Thanks for this suggestion. We have re-drawn the figures in this revised manuscript (see Figs. 1j, 2e, 4d-f and 5g). More information has been provided in detail.

Reviewers' comments:

Reviewer #1 (Remarks to the Author):

I thank the authors efforts on addressing most of my comments. However, it look like the authors are too hurry in pushing the important results, thus do not consider referee's comments seriously. As the H doping acts as cruciul role through whole discussion, how can a simple FT-IR measurment prove all? I agree the Ni-H doping peak in FT-IR presented in this paper is similar to that in cited referece, but it is really not convincing evidence. It is main reason that hinders my recommendation. A more precise approach is suggeted to determine the real H-Ni, such as solid-staet H NMR or EXAFS. Moreover, since both methanol and H₂O might be involved with H₂ generation, and the mixture is also able to generate H₂ under light without photocatalyst ((Sci. Rep. 5, 13475 (2015))), other hole scavengers, such as DETA, Na₂SO₃ seems to be better than the methanol for photocatalytic H₂ evaluation.

Reviewer #2 (Remarks to the Author):

Moderate corrections were made.

Reviewer #3 (Remarks to the Author):

The point of the present paper is whether 2DPA functions as photoatalysts with Ni-H structure, and the paper has been revised on the basis of new experimental findings, in which the stability of Ni-H structure was demonstrated. With photocatalysis, however, the following two points should be clarified to further verify the photocatalysis of 2DPA.

(1) The authors have proposed the mechanism of hydrogen evolution from CH₃OH. Together with a large amount of hydrogen, CO₂ is also produced in this mechanism. To confirm the photocatalysis, the amount of CO₂ should be analyzed and compared with the amount of H₂.

(2) The paper says that there is no O₂ evolution in the sacrificial reaction of AgNO₃ aqueous solution by light irradiation. Possibly no deposition of Ag on the surface. If the photocatalysis is concerned, these results suggest that the recombination of photoexcited electrons and holes are at least remarkably fast, indicating that the electrons are able to transfer to H⁺ only when the holes are consumed off by the reaction with CH₃OH. When the photo-irradiated sacrificial reaction of AgNO₃ was examined in CH₃OH/H₂O (under the conditions without the thermal reduction of Ag⁺), the view, that is, photocatalysis, is though to be confirmed if the Ag deposition (electron transfer) can be observed.

Response to Referees

For Reviewer #1

Comments #1. "I thank the authors efforts on addressing most of my comments. However, it look like the authors are too hurry in pushing the important results, thus do not consider referee's comments seriously. As the H doping acts as crucial role through whole discussion, how can a simple FT-IR measurment prove all? I agree the Ni-H doping peak in FT-IR presented in this paper is similar to that in cited referece, but it is really not convincing evidence. It is main reason that hinders my recommendation. A more precise approach is suggeted to determine the real H-Ni, such as solid-staet H NMR or EXAFS."

Our Response. We accepted the referee's suggestion and the solid-state ^1H NMR measurements have been carried out in this revised manuscript. The ^1H NMR spectra of 2DA and 2DPA are shown in Fig. 1g. Clearly, all of the peaks are broad, which should be due to the amorphous nature of 2DA and 2DPA and the resulting various chemical environments for H atoms (*Adv. Funct. Mater.* **23**, 5444-5450 (2013)). Three common peaks can be found for 2DA and 2DPA (labeled as A, B and C). They can be respectively assigned to the terminal Ni-OH, internal Ni-OH and surface adsorbed -OH groups (*J. Chem. Soc., Faraday Trans.* **92**, 2791-2798 (1996)). Since surface Ni atoms of NiO nanoflakes are generally under-coordinated, terminal Ni-OH groups form more easily, and peak A is much larger than peak B. In contrast to 2DA, 2DPA exhibits two additional resonances at chemical shifts of 3.4 and -8.4 ppm. They should be associated with the bound H atoms on Ni metal atoms (*Angew. Chem. Int. Ed.* **53**, 1081-1086 (2014)). In detail, they are respectively terminal and internal Ni-H groups (*J. Am. Chem. Soc.* **139**, 16720-16731 (2017); *J. Am. Chem. Soc.* **126**, 4566-4580 (1964)).

Accordingly, internal Ni-H groups are much more than terminal ones, suggesting that H atoms have been doped into the NiO lattices. At this point, the presence of Ni-H bonds in 2DPA has been demonstrated.

Comments #2. "Moreover, since both methanol and H₂O might be involved with H₂ generation, and the mixture is also able to generate H₂ under light without photocatalyst ((*Sci. Rep.* **5**, 13475 (2015))), other hole scavengers, such as DETA, Na₂SO₃ seems to be better than the methanol for photocatalytic H₂ evaluation."

Our Response. We accepted the referee's suggestion and H-free hole scavenger ($\text{Na}_2\text{S}/\text{Na}_2\text{SO}_3$) has been used in this revised manuscript. Photocatalyst (100.0 mg) was dispersed in 100 mL $\text{Na}_2\text{S}/\text{Na}_2\text{SO}_3$ aqueous solution (Na_2S 0.35 M, Na_2SO_3 0.25 M) upon AM 1.5 irradiation. Typical time courses of H_2 evolution are shown in Supplementary Fig. 13. Both 2DA and 2DPA still show favorable photocatalytic activities with H_2 evolution rates of 48.90 and 852.54 $\mu\text{mol}/h$.

These results above confirm that 2DA and 2DPA are efficient photocatalysts to realize H_2 evolution from water.

For Reviewer #3

Comments #1. "The authors have proposed the mechanism of hydrogen evolution from CH_3OH . Together with a large amount of hydrogen, CO_2 is also produced in this mechanism. To confirm the photocatalysis, the amount of CO_2 should be analyzed and compared with the amount of H_2 ."

Our Response. We accepted the referee's suggestion and we have carried out the measurement of CO_2 in our case. In the revised manuscript, CO_2 production was detected as shown in Supplementary Fig. 11, which showing that CO_2 was released over 2DA and 2DPA during photocatalytic H_2 evolution. The CO_2 production rates are 7.39 and 132.31 $\mu\text{mol}/h$ respectively for 2DA and 2DPA. The corresponding ratios of evolved H_2 to CO_2 are 7.04 and 7.62. The value is much larger than that in the proposed methanol reforming reaction (3.0). In addition, to achieve an optimal photocatalytic performance in the methanol reforming reaction, a large methanol concentration, even larger than 95 vol.%, is required. In this work, both 2DA and 2DPA obtain the optimal H_2 evolution rate when the methanol concentration is low (20%).

These results above suggest that the photocatalytic behavior is not just a methanol reforming process, and H atoms in H_2O should make a great contribution on the produced H_2 .

Comments #2. "The paper says that there is no O_2 evolution in the sacrificial reaction of AgNO_3 aqueous solution by light irradiation. Possibly no deposition of Ag on the surface. If the photocatalysis is concerned, these results suggest that the recombination of photoexcited electrons and holes are at least remarkably fast,

indicating that the electrons are able to transfer to H⁺ only when the holes are consumed off by the reaction with CH₃OH. When the photo-irradiated sacrificial reaction of AgNO₃ was examined in CH₃OH/H₂O (under the conditions without the thermal reduction of Ag⁺), the view, that is, photocatalysis, is thought to be confirmed if the Ag deposition (electron transfer) can be observed."

Our Response. We appreciated these suggestions and we have done the relevant experiments in the revised manuscript according to these suggestions.

Considering the fact that H₂ and O₂ cannot be produced in pure water without hole scavengers over 2DA and 2DPA, photocatalytic O₂ evolution, the other half-reaction of overall water splitting, was performed in AgNO₃ aqueous solution (0.03 M, 100 mL) using Ag⁺ as the electron scavengers at 4°C. Unfortunately, no O₂ signal can be detected. Supplementary Figs. 14a and b show TEM images of the two samples after the AgNO₃ sacrificial reactions (labeled as 2DA-AgNO₃ and 2DPA-AgNO₃). Obviously, Ag nanoparticles (black dots) have been deposited onto the surfaces of the photocatalysts, which is further demonstrated by the XRD patterns in Supplementary Fig. 15. The peaks at 38.4, 44.1 and 66.3° can be ascribed to the cubic Ag phase (JCPDS 04-0783). 2DPA-AgNO₃ possesses more Ag nanoparticles due to its stronger light harvesting and carrier generation ability (Fig. 2c). Therefore, these results confirm the electron transfer behavior during the photocatalytic process. Since Ag⁺ is more easily reducible than H⁺, H₂ cannot be produced in pure water without methanol (*J. Mater. Chem. A* **3**, 2485-2534 (2015)).

Moreover, the AgNO₃ sacrificial reactions were conducted in 20 vol.% methanol aqueous solution. It can be clearly found that more Ag nanoparticles are deposited (Supplementary Figs. 14c-d). The Ag amount for 2DPA-AgNO₃-CH₃OH is still larger than that for 2DA-AgNO₃-CH₃OH (Supplementary Fig. 15). These results indicate that the electron transfer can be enhanced by adding methanol into AgNO₃ aqueous solution. That would be the reason for the requirement of hole scavengers in the photocatalytic H₂ evolution process. As shown in Supplementary Fig. 12, H₂ evolution can still be realized when inadequate methanol (10 vol.%) was added.

In spite of the electron transfer behavior, photocatalytic O₂ evolution cannot be achieved over 2DA and 2DPA. It is discussed above that NiO has been widely used as H₂ evolution catalysts instead of O₂ evolution catalysts due to its strong interaction with H atoms (*Chem. Soc. Rev.* **38**, 253-278 (2009)). The lack of active sites will lead

to the failure in O₂ evolution. Moreover, it is known that photocatalytic O₂ evolution is a four-electron process requiring a large overpotential (*Science* **347**, 970-974 (2015)). Fig. 2e indicates that the VBMs of 2DA and 2DPA are not much lower than the O₂ evolution level (1.23 V vs. NHE). Their O₂ evolution sluggishness may be due to the kinetic limitation of the oxidation process, as well.

Accordingly, these results that Ag deposition can be observed in AgNO₃/H₂O and AgNO₃/CH₃OH/H₂O confirm the electron transfer behavior during photocatalytic H₂ evolution in our case.

Reviewers' comments:

Reviewer #1 (Remarks to the Author):

This paper can be accepted in its present form. congratulations!

Reviewer #3 (Remarks to the Author):

With the additional experiments, a large amount of CO₂ from CH₃OH was observed for 2DA and 2DPA by light irradiation, which clearly verifies the photocatalytic evolution of H₂ from CH₃OH/H₂O solution. Furthermore, the TEM observation showed that light irradiation of 2DA and 2DPA in Ag(NO₃) dissolved in H₂O and CH₃OH/H₂O produced Ag aggregates on the surfaces. The Ag deposition occurred largely for CH₃OH/H₂O than H₂O in line with the activity of hydrogen evolution. These results clearly demonstrate the role of photoexcited electrons.

The problem is the performance of holes. No oxygen evolution was observed from the reaction systems of Ag(NO₃)/H₂O and Ag(NO₃)/CH₃OH/H₂O. Although the paper insists that H₂O in CH₃OH contributes to the hydrogen evolution (Page14, line 288), there is no observation with oxygen evolution (where does the oxygen in H₂O go?). The authors explain that the VBM level is close to O₂/H₂O level (Page16), but this is not acceptable, because the band gaps of 2DA and 2DPA are as large as 2.99-3.24 V(Supporting Information Fig. 9), and the maximum in the SPV spectra appears at around of 400- 530 nm(Fig. 4).

There is no doubt that the present study has been nicely carried out using many sophisticated instruments and detailed analysis under controlled experimental conditions, and some interesting phenomena have been observed. However, it is certain that it seems difficult to understand the very core of the treated materials, possibly due to the complex structures and components of the materials. Namely, in spite of tremendously large amount hydrogen evolution, there is little oxygen production. This raises question with the availability of the present materials as photocatalysts from a viewpoint of solar H₂ from H₂O, and is concluded to be unsuitable for Nature Communications.

Response to Reviewer 3

For Comments #1. "With the additional experiments, a large amount of CO₂ from CH₃OH was observed for 2DA and 2DPA by light irradiation, which clearly verifies the photocatalytic evolution of H₂ from CH₃OH/H₂O solution. Furthermore, the TEM observation showed that light irradiation of 2DA and 2DPA in Ag(NO₃) dissolved in H₂O and CH₃OH/H₂O produced Ag aggregates on the surfaces. The Ag deposition occurred largely for CH₃OH/H₂O than H₂O in line with the activity of hydrogen evolution. These results clearly demonstrate the role of photoexcited electrons."

Our Response. Thanks a lot for these positive comments about our experimental results.

For Comments #2. "The problem is the performance of holes. No oxygen evolution was observed from the reaction systems of Ag(NO₃)/H₂O and Ag(NO₃)/CH₃OH/H₂O. Although the paper insists that H₂O in CH₃OH contributes to the hydrogen evolution (Page14, line 288), there is no observation with oxygen evolution (where does the oxygen in H₂O go?). The authors explain that the VBM level is close to O₂/H₂O level (Page16), but this is not acceptable, because the band gaps of 2DA and 2DPA are as large as 2.99-3.24 V(Supporting Information Fig. 9), and the maximum in the SPV spectra appears at around of 400- 530 nm(Fig. 4)."

Our Response. Thanks for the valuable remarks from Reviewer #3. According to the suggestions of Reviewer #3, we have carried out new experiments and detailed analysis to pursue the issues (performances of holes) mentioned by the referee, and the results were as follows.

(1) Reactive oxygen species (ROS) evolution route in AgNO₃ solution

The detection of $\cdot\text{O}_2^-$, $^1\text{O}_2$ and $\cdot\text{OH}$ was performed by the ESR-spin trapping technique using BMPO, TEMP and DMPO as the trapping agents, respectively. As shown in Supplementary Fig. 16a, in the absence of a photocatalyst, no signal could be detected (labeled as blank). When photocatalyst powders were added into the system, a four-line spectrum with the relative intensities of 1:1:1:1 formed, which is the characteristic spectrum for the BMPO- $\cdot\text{O}_2^-$ adducts (*J. Catal.* 320, 97-105 (2014)). Similarly, four characteristic peaks with the intensity ratios of 1:2:2:1 appeared when DMPO was used as the trapping agent, shown in Supplementary Fig. 16b. These peaks should be attributed to the DMPO- $\cdot\text{OH}$ adducts (*Chem. Eng. J.* 170, 353-362 (2011)). On the contrary, no TEMP- $^1\text{O}_2$ adducts was detected (triplet signals), indicating the absence of $^1\text{O}_2$ radicals in the process (*Catal. Sci. Technol.* 8, 2186-2194 (2018)). The generation of H_2O_2 was determined by the coloration method, and the absorbance spectra are shown in Supplementary Fig. 16c. The spectra show a characteristic peak with the wavelength of 439 nm, demonstrating the existence of H_2O_2 in the reaction solutions (*J. Mater. Chem. A* 5, 19800-19807 (2017)).

Evidently, $\cdot\text{O}_2^-$ radicals, H_2O_2 and $\cdot\text{OH}$ radicals can be detected in the AgNO₃ solution containing 2DA or 2DPA upon irradiation, whereas $^1\text{O}_2$ was absent.

Compared with 2DA, 2DPA owns stronger signals for the detection of $\cdot\text{O}_2^-$, H_2O_2 and $\cdot\text{OH}$, which should be due to its stronger light harvesting ability. Considering that the oxidation potentials of H_2O to $\cdot\text{OH}$ (+2.31 V vs. NHE) and OH^- to $\cdot\text{OH}$ (+1.99 V vs. NHE) are more positive than the VB tops of 2DA (+1.78 V vs. NHE) and 2DPA (+1.85 V vs. NHE), we believe that $\cdot\text{OH}$ radicals should not be generated through the direct oxidation of H_2O or OH^- . Namely, $\cdot\text{OH}$ should not be produced first. Two routes for the ROS evolution may be involved (**Route A: $\text{H}_2\text{O} \rightarrow \cdot\text{O}_2^- \rightarrow \text{H}_2\text{O}_2 \rightarrow \cdot\text{OH}$** ;

Route B: $\text{H}_2\text{O} \rightarrow \text{H}_2\text{O}_2 \rightarrow \cdot\text{O}_2^- / \cdot\text{OH}$, which can be described by the following equations.

Route A (*Nat. Commun.* 9, 1397 (2018); *J. Mater. Chem. A* 3, 7649-7658 (2015)):

Route B (*Chem. Rev.* 117, 11302-11336 (2017); *Environ. Sci. Technol.* 45, 3027-3033 (2011)):

The oxidation potential of H_2O to H_2O_2 is reported as +1.78 V vs. NHE. Given that the VB tops of 2DA and 2DPA are very close to this value, we consider that **Route A may be more applicable to this case**, which has also been reported by Tian *et al.* (*Nat. Commun.* 9, 1397 (2018)). Certainly, one may consider that in this process, another route (Route C: $\text{H}_2\text{O} \rightarrow \text{O}_2 \rightarrow \cdot\text{O}_2^- \rightarrow \text{H}_2\text{O}_2 \rightarrow \cdot\text{OH}$) should be involved, in which O_2 is generated first. However, in one hand, O_2 was not detected. In the other hand, the oxidation of H_2O to O_2 is a four-electron process requiring a large overpotential and more active sites to realize the O-O coupling (*Science* 347, 970-974 (2015)). Therefore, Route C may be less possible in our case.

The possible reaction route for the system containing AgNO_3 and photocatalyst (2DA or 2DPA) upon irradiation is illustrated in Supplementary Fig. 17. Upon irradiation,

electrons and holes are generated (❶). Parts of them are recombined unavoidably (❷). Due to the strong electron-harvesting ability of Ag^+ , the electrons are consumed by the Ag^+ ions quickly, and Ag nanoparticles are deposited onto the surface of the photocatalyst (❸). The remaining holes oxidize H_2O to generate $\cdot\text{O}_2^-$ radicals (❹). H_2O_2 is further produced through the reduction of $\cdot\text{O}_2^-$ by the electrons (❺). Meanwhile, the decomposition of H_2O_2 into $\cdot\text{OH}$ and the dimerization of $\cdot\text{OH}$ proceed (❻). The reaction between $\cdot\text{OH}$ and H^+ to form H_2O again may take place, as well (❼). Among the ROS involved above, H_2O_2 is the most stable (*J. Phys. Chem. Lett.* 6, 958-963 (2015)). Therefore, **the whole process can be described simply by the equation**

Further, when methanol is added, the photo-generated holes can be consumed more quickly. Recombination between electrons and holes (❷) is suppressed, and more electrons are left for the Ag^+ reduction (❸). Therefore, more Ag nanoparticles are formed.

(2) ROS evolution route in pure water

Photocatalytic overall water splitting over 2DA and 2DPA was evaluated, as well. Considering the fact that H_2 and O_2 cannot be produced in pure water without any scavengers (methanol or AgNO_3), the ROS were also monitored. As shown in Supplementary Fig. 18, $\cdot\text{O}_2^-$ and $\cdot\text{OH}$ radicals can still be found. However, H_2O_2 disappeared.

Ag^+ is a well-known electron scavenger that can consume the photo-generated electrons timely and retard the undesired carrier recombination (*J. Phys. Chem. Lett.* 4, 3479-3483 (2013)). At this point, ROS generation can proceed more smoothly. When

the photocatalytic tests are conducted in pure water without Ag^+ , the consumption of electrons is weakened since H^+ ions are less easily reducible than Ag^+ ions (*J. Mater. Chem. A* 3, 2485-2534 (2015)). Actually, electron consumption and hole consumption are interactional. Although some of the holes can be used for producing ROS ($\cdot\text{O}_2^-$ and $\cdot\text{OH}$), they cannot be consumed quickly due to the excess of electrons. As shown in Supplementary Fig. 18, the signal to noise ratios of the ESR spectra are much smaller than those in Supplementary Fig. 16. The accumulation of electrons and holes in the photocatalysts will enhance the carrier recombination.

We consider that the ROS evolution route in pure water may be similar to that in AgNO_3 solution ($\text{H}_2\text{O} \rightarrow \cdot\text{O}_2^- \rightarrow \text{H}_2\text{O}_2 \rightarrow \cdot\text{OH}$). The absence of H_2O_2 in this case may be due to the decomposition of H_2O_2 into $\cdot\text{OH}$. In spite of the electron transfer behavior, excess $\cdot\text{OH}$ will be reduced through the following reaction

Therefore, O_2 was not produced, and H_2 evolution was hindered, as well. Proton reduction and water oxidation cannot take place simultaneously (Supplementary Fig. 19).

Accordingly, possible ROS evolution routes involving the photo-generated electrons and holes in AgNO_3 solution or pure water have been proposed based on the ROS detection and band structure analysis in this revised manuscript.

Overall, additional experiments and detailed analysis in this revised manuscript have demonstrated the existence of abundant ROS and revealed the evolution of the photo-generated holes and O atoms in H_2O . Related experimental results and discussions have been added in the revised manuscript and Supplementary Information.

Additionally, according to the reviewer's remark and new experiments, we accepted

the suggestion of the referee, and we have removed the previous discussions about the correlation between the band structures and the O₂ evolution sluggishness in this revised manuscript.

For comments #3. "There is no doubt that the present study has been nicely carried out using many sophisticated instruments and detailed analysis under controlled experimental conditions, and some interesting phenomena have been observed.

Our response. Thanks a lot for these positive comments about our experimental results and detailed analysis.

For comments #4. "However, it is certain that it seems difficult to understand the very core of the treated materials, possibly due to the complex structures and components of the materials. Namely, in spite of tremendously large amount hydrogen evolution, there is little oxygen production. This raises question with the availability of the present materials as photocatalysts from a viewpoint of solar H₂ from H₂O, and is concluded to be unsuitable for Nature Communications."

Our response. First of all, by adding experimental studies and detailed analysis, we have proposed the more reasonable photocatalytic mechanism including performances of electrons and holes in this revised manuscript. Importantly, new experiments have demonstrated the existence of abundant ROS and revealed the evolution of the photo-generated holes and O atoms in H₂O in our case. Now, we believe that people can easily and clearly understand that 2DA and 2DPA are efficient photocatalysts to realize H₂ evolution from H₂O in this revised manuscript.

Secondly, photocatalysts have so far focused on crystals since Fujishima and Honda realized water splitting over a TiO₂ single crystal in 1972 (*Nature* 238, 37-38 (1972)),

whereas amorphous materials are usually evaluated to be photocatalytically inactive due to the self-trapping of tail states. On the other hand, NiO crystals fail to act as a photocatalyst since it is intrinsically hole-doped with a very low ratio of free carriers to trapped holes. In this work, for the first time, we demonstrate that 2DA can act as an efficient and robust photocatalyst for solar H₂ evolution. Further, the antenna effect of SPR can be further introduced into 2DPA to construct an incorporate antenna-reactor structure. The solar H₂ evolution rate is improved by a factor of 19.4 through the SPR-mediated charge releasing. To the best of our knowledge, it is the first time to construct an incorporate antenna-reactor plasmonic structure for solar H₂ evolution. Thus, these findings actually open a door to applications of 2D amorphous materials as advanced photocatalysts.

Finally, considering the originality and systematic characteristics of this work, we are sure that this revised manuscript should be suitable for the publication in *Nature Communications*.

REVIEWERS' COMMENTS:

Reviewer #3 (Remarks to the Author):

The reactivity of hole in the system of AgNO₃ aqueous solution on 2DA and 2DPA was investigated by ESR and the coloration methods. The results show that the hole produces H₂O₂, together with superoxide and hydroxyl radicals. The whole picture of photocatalysis, i.e., the action of electrons and holes, is obtained.

While I still have concerns over the photocatalysis of the materials, I don't think that they should be a reason to further hold back its publication after the following comments are clarified.

- (1) The amount of the radicals is possibly extremely lower because of highly sensitive EPR measurements, but what about the amount of H₂O₂ ? Is it comparable with the amount of Ag deposited?
- (2) Do you see the production of neither oxygen radicals nor H₂O₂ in the reaction system of AgNO₃ methanol solution?
- (3) To show the energy level of the reactions concerned (CH₃OH to CO₂, Ag⁺ to Ag, H₂O to H₂O₂ etc), together the conduction and valence levels of 2DPA, would be preferable for understanding photocatalytic scheme.

Response to Referee #3

General Comments. "The reactivity of hole in the system of AgNO₃ aqueous solution on 2DA and 2DPA was investigated by ESR and the coloration methods. The results show that the hole produces H₂O₂, together with superoxide and hydroxyl radicals. The whole picture of photocatalysis, i.e., the action of electrons and holes, is obtained. While I still have concerns over the photocatalysis of the materials, I don't think that they should be a reason to further hold back its publication after the following comments are clarified."

Our Response. Thanks a lot for these positive comments about our work.

Comments #1. "The amount of the radicals is possibly extremely lower because of highly sensitive EPR measurements, but what about the amount of H₂O₂? Is it comparable with the amount of Ag deposited?"

Our Response. Thanks for this valuable remark. According to the calibration curve in Supplementary Fig. 19, H₂O₂ concentrations are respectively determined as 47.07 and 260.08 μmol L⁻¹ for 2DA-AgNO₃ and 2DPA-AgNO₃. The H₂O₂ amounts are respectively 4.71 and 26.01 μmol. Inductively, coupled plasma-mass spectrometry (ICP-MS) was employed to analyze the Ag amounts. They are respectively 1.52 mg (14.07 μmol) and 6.41 mg (59.35 μmol). H₂O₂ amounts are comparable with the Ag amounts. In detail, the Ag amounts are approximately twice the H₂O₂ amounts. It is accord with the proposed reaction route ($2\text{Ag}^+ + 2\text{H}_2\text{O} \rightarrow 2\text{Ag} + \text{H}_2\text{O}_2 + 2\text{H}^+$). These results demonstrate the proposed reaction route.

See Supplementary Figure 19.

Comments #2. "Do you see the production of neither oxygen radicals nor H₂O₂ in the reaction system of AgNO₃ methanol solution?"

Our Response. When methanol was added, neither oxygen radicals nor H₂O₂ can be detected. This result indicates that the photo-generated holes can be consumed by methanol quickly.

See Supplementary Note 7.

Comments #3. "To show the energy level of the reactions concerned (CH₃OH to CO₂, Ag⁺ to Ag, H₂O to H₂O₂ etc), together the conduction and valence levels of 2DPA, would be preferable for understanding photocatalytic scheme."

Our response. Thanks for this suggestion. Schematic representation of the band structures for 2DA and 2DPA as well as the involved redox potentials has been added in Supplementary Information as Supplementary Figure 17.